# HyPlaneHead: Rethinking Tri-plane-like Representations in Full-Head Image Synthesis

**Heyuan Li**[*]
SSE, CUHK (Shenzhen)
heyuanli@link.cuhk.edu.cn

**Kenkun Liu**
SSE, CUHK (Shenzhen)
kenkunliu@link.cuhk.edu.cn

**Lingteng Qiu**[†]
Tongyi Lab, Alibaba Inc.
qiulingteng.qlt@alibaba-inc.com

**Qi Zuo**
Tongyi Lab, Alibaba Inc.
muyuan.zq@alibaba-inc.com

**Keru Zheng**
SSE, CUHK (Shenzhen)
keruzheng@link.cuhk.edu.cn

**Zilong Dong**
Tongyi Lab, Alibaba Inc.
list.dzl@alibaba-inc.com

**Xiaoguang Han**[‡]
SSE, CUHK (Shenzhen)    FNii-Shenzhen
Guangdong Provincial Key Laboratory of Future Networks of Intelligence
hanxiaoguang@cuhk.edu.cn

## Abstract

Tri-plane-like representations have been widely adopted in 3D-aware GANs for head image synthesis and other 3D object/scene modeling tasks due to their efficiency. However, querying features via Cartesian coordinate projection often leads to feature entanglement, which results in mirroring artifacts. A recent work, SphereHead, attempted to address this issue by introducing spherical tri-planes based on a spherical coordinate system. While it successfully mitigates feature entanglement, SphereHead suffers from uneven mapping between the square feature maps and the spherical planes, leading to inefficient feature map utilization during rendering and difficulties in generating fine image details. Moreover, both tri-plane and spherical tri-plane representations share a subtle yet persistent issue: feature penetration across convolutional channels can cause interference between planes, particularly when one plane dominates the others (see fig. 1). These challenges collectively prevent tri-plane-based methods from reaching their full potential. In this paper, we systematically analyze these problems for the first time and propose innovative solutions to address them. Specifically, we introduce a novel hybrid-plane (hy-plane for short) representation that combines the strengths of both planar and spherical planes while avoiding their respective drawbacks. We further enhance the spherical plane by replacing the conventional theta-phi warping with a novel near-equal-area warping strategy, which maximizes the effective utilization of the square feature map. In addition, our generator synthesizes a single-channel unified feature map instead of multiple feature maps in separate channels, thereby effectively eliminating feature penetration. With a series of technical improvements, our hy-plane representation enables our method, HyPlaneHead, to achieve state-of-the-art performance in full-head image synthesis.

---

[*]Work done during an internship at Tongyi Lab, Alibaba Inc.
[†]Project lead.
[‡]Corresponding author.

39th Conference on Neural Information Processing Systems (NeurIPS 2025).

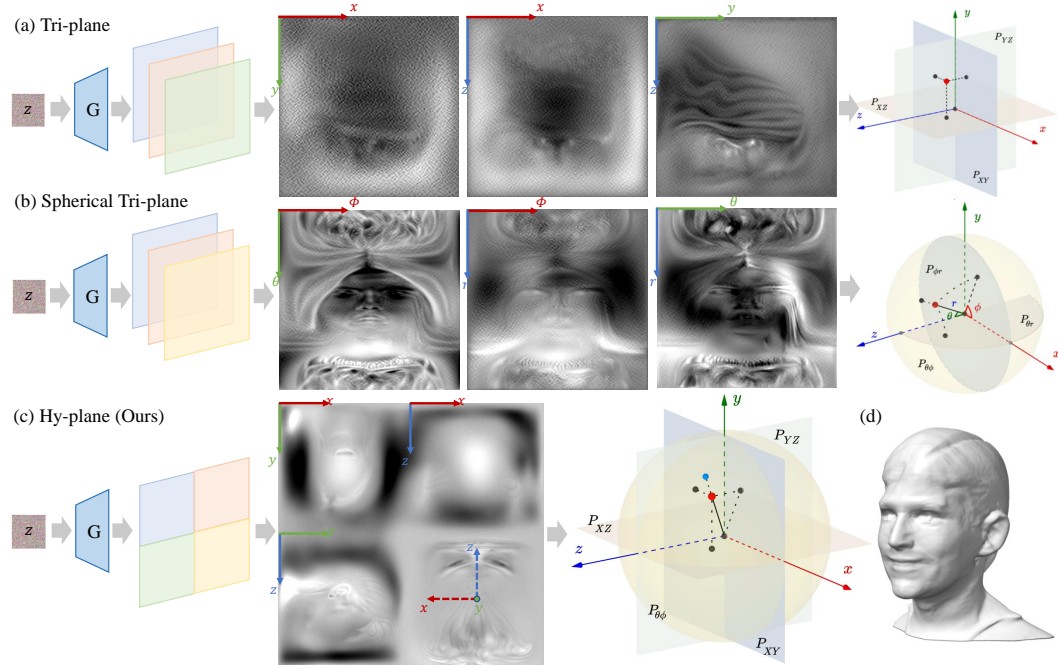

Figure 1: Figure (a, b, c) respectively illustrate the feature map visualizations and geometric structures of the tri-plane, spherical tri-plane, and our proposed hy-plane representation which integrates both planar and spherical planes. Figure (d) shows a head geometry model for defining the coordinate system. Note that in (a, b), the dominant planes ($P_{YZ}$ for tri-plane and $P_{\theta\phi}$ for spherical tri-plane) cause significant inter-channel feature penetration into the other two planes (best viewed when zoomed in), thereby limiting the model's expressiveness. In contrast, (c) resolves this issue entirely by employing a unify-split strategy, where all feature maps are generated within a single channel. As a result, each plane effectively learns its corresponding information without interference from other planes.

# 1 Introduction

Photorealistic full-head synthesis Zhuang et al. (2022); Park et al. (2021); Canela et al. (2023); He et al. (2024); Doukas et al. (2021) stands as a cornerstone technology for emerging applications in augmented/virtual reality avatars, immersive telepresence systems, and next-generation digital content creation. While modern 2D generative adversarial networks (GANs) Goodfellow et al. (2020); Radford et al. (2015); Mao et al. (2017); Gulrajani et al. (2017); Zhou et al. (2021); Kang et al. (2023) achieve remarkable image quality in frontal face generation, their fundamental limitation in 3D scene modeling becomes apparent when synthesizing head images under arbitrary viewpoints.

Recent advancements in 3D-aware GANs Schwarz et al. (2020); Deng et al. (2022); Xue et al. (2022); Nguyen-Phuoc et al. (2019); Chan et al. (2021); Shi et al. (2021); Chan et al. (2022); An et al. (2023); Li et al. (2024) have tackled this challenge by leveraging neural implicit representations, enabling view-consistent synthesis while maintaining photorealistic quality. Among these methods, the pioneering work EG3D Chan et al. (2022) employs a tri-plane structure to represent human heads or other 3D objects. The tri-plane representation Gao et al. (2022); Shue et al. (2023); Zou et al. (2024); Wang et al. (2023b); Hong et al. (2023); Gupta et al. (2023); Zuo et al. (2023); Wu et al. (2024a); Zuo et al. (2024) efficiently captures symmetrical regions because two 3D points that are symmetric with respect to a feature plane will query the same feature on the plane via Cartesian coordinate projection. However, this inherent coupling of features becomes problematic in asymmetrical areas, leading to mirroring artifacts. As shown in fig. 2 (a, b), a typical example in full-head synthesis is that the back-view of the head shares the same features on the $P_{XY}$ plane as the front-view face, resulting in noticeable fake face artifacts on the back of the head. While PanoHead An et al. (2023) mitigates this issue by augmenting each plane with additional parallel planes, its tri-grid representation does not fundamentally resolve the problem, as it still inherits the same geometric and projection limitations of the Cartesian coordinate system.

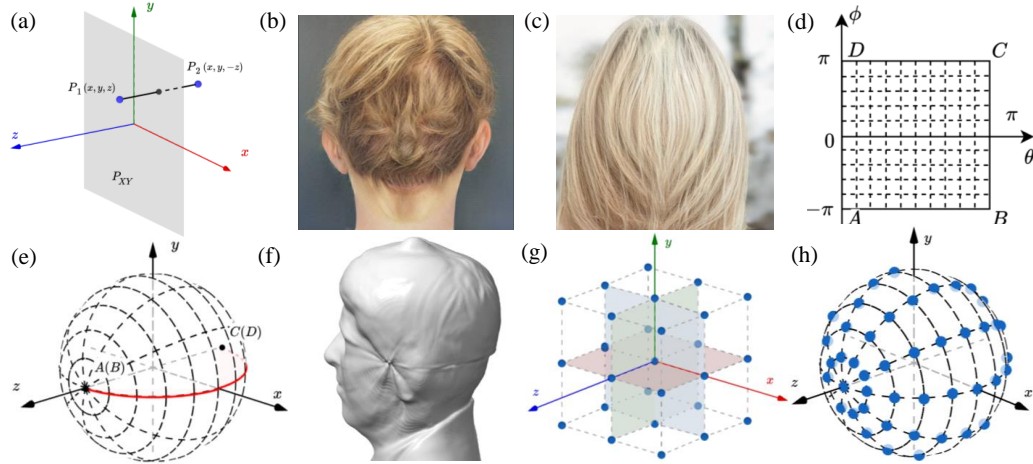

Figure 2: In the tri-plane representation, (a) the feature entanglement issue results in mirroring artifacts, where (b) the back of the head incorrectly exhibits front-face attributes, or (c) the hair's texture and shape display an unnaturally high degree of left-right symmetry. In the spherical tri-plane representation, (d, e) the non-equal-area warping caused by mapping a square to a sphere using $(\theta, \phi)$ coordinates introduces (f) artifacts in the seam and polar regions, as well as uneven spatial feature distribution after warping. By contrast, (g) the tri-plane representation exhibits even spatial feature distribution, whereas (h) the spherical tri-plane representation shows uneven distribution, with features being overly dense in the polar regions and sparse near the equator.

A recent work, SphereHead Li et al. (2024), creatively addresses the mirroring issue by introducing a spherical tri-plane representation that projects features in a spherical coordinate system. However, this approach introduces new challenges. First, it fails to leverage symmetry, which is prevalent in real-world objects. Second, the mapping from the square feature map to the spherical plane $P_{\theta\phi}$ involves a non-equal-area projection. Specifically, as illustrated in fig. 2 (c-f), after this mapping, features are sparsest near the equator and densest at the poles. This uneven distribution results in inefficient feature map utilization when rendering 2D images, reducing the model's ability to capture fine details. Moreover, referring to fig. 2 (d), a single spherical tri-plane can produce artifacts in the seam region due to the numerical discontinuity of $P_{\theta\phi}$ at $\phi = -\pi$ and $\phi = \pi$ in the spherical coordinate system. Although SphereHead mitigates this issue by incorporating an additional orthogonal spherical tri-plane, this solution complicates the model and introduces parameter redundancy. Worse still, the least expressive equatorial region of one sphere are used to cover the most expressive polar regions of the other, further diminishing the overall expressiveness of the representation.

Besides, we are the first to observe that a subtle yet persistent issue exists in both tri-plane and spherical tri-plane representations: feature penetration across convolutional channels can lead to interference between feature planes, particularly when one plane dominates the others. This issue arises because, unlike RGB images where channels are spatially aligned in 2D, each feature plane has a unique distribution, resulting in significantly different spatial meaning and values at the same uv position. In convolutional layers, however, all output channels at a given uv position are computed using the same input values. Ideally, the network is supposed to learn appropriate kernels to separate information for different planes. Yet, this is particularly challenging for 3D-aware GANs, as they are trained on 2D images without direct supervision on feature maps. Consequently, feature penetration often manifests visibly across planes, as shown in fig. 1. Although visible feature penetration gradually diminishes as training progresses, the issue itself remains difficult to fully resolve, subtly limiting the model's expressiveness and causing seemingly inexplicable artifacts.

In this paper, we introduce a simple yet effective **unify-split strategy** that generates a single-channel feature map and then splits it into multiple feature planes, instead of using different output channels to generate different feature planes. This approach completely eliminates the issue of feature penetration between channels. Building on this, we propose a novel **hybrid-plane (hy-plane) representation** that integrates both planar and spherical feature planes, as illustrated in fig. 1 (c). This design leverages the strengths of both tri-plane and spherical tri-plane representations while mitigating their respective limitations. Specifically, the hy-plane representation automatically learns symmetrical

features using planar planes and captures anisotropic features through the spherical plane. This approach avoids mirroring artifacts while ensuring a uniformly high feature density throughout 3D rendering. Furthermore, to optimize the mapping from the square feature map to the spherical plane, we employ Lambert azimuthal equal-area projection Öztürk (2024) combined with elliptical grid mapping Fong (2019). These techniques maximize the utilization of the square feature map and eliminate the seam artifacts. We also explore several variant models to further enhance performance. For instance, we increase the area proportion of the spherical plane to boost its expressive power and propose a dual-plane-dual-sphere variant to fully resolve polar artifacts. These innovations collectively contribute to the robustness and versatility of our hy-plane representation.

Building on the aforementioned technical advancements, the novel hy-plane representation enables our **HyPlaneHead** model to achieve state-of-the-art performance in full-head image synthesis, delivering high-quality results with significantly fewer artifacts compared to existing 3D-aware GAN methods Schwarz et al. (2020); Deng et al. (2022); Xue et al. (2022); Nguyen-Phuoc et al. (2019); Chan et al. (2021); Shi et al. (2021); Chan et al. (2022); An et al. (2023); Li et al. (2024). In summary, our main contributions are as follows:

- We conduct an in-depth analysis of the limitations inherent in tri-plane-like representations used in 3D-aware GANs. Based on this understanding, we introduce the *hy-plane representation*, which combines the strengths of both planar and spherical planes while addressing their respective drawbacks.

- To achieve seamless integration of planar and spherical planes, we propose a series of technical innovations, including *unify-split strategy*, a novel *near-equal-area warping* method, *area-biased splitting*, and exploration of *alternative combination strategies*.

- Through comprehensive experiments, we validate the effectiveness of our proposed representation. Our *HyPlaneHead* model achieves state-of-the-art performance for full-head image synthesis, demonstrating superior quality and reduced artifacts.

## 2   Related Work

**3D Morphable Head Representations.** Traditional approaches for representing 3D faces with diverse shapes and appearances rely on 3D Morphable Models (3DMM) Blanz and Vetter (1999); Paysan et al. (2009), with FLAME Li et al. (2017) extending this framework to full head modeling. However, the coarse geometric details provided by 3DMMs have motivated numerous works to combine them with implicit neural representations, such as NeRF Canela et al. (2023); Zanfir et al. (2022); Zheng et al. (2022); Gafni et al. (2021); Guo et al. (2021); Park et al. (2021); Wu et al. (2023a); Yenamandra et al. (2021); Zhang et al. (2023); Zhuang et al. (2022). While volume-based rendering techniques have significantly enhanced the capabilities of 3DMM-based models, their inherent topological constraints limit the expressiveness of implicit representations, particularly in capturing fine details like hair and wrinkles. Consequently, recent 3D-aware generative models have shifted towards directly synthesizing implicit neural fields of heads without relying on 3DMM priors.

**Generative Neural Head Representations.** Emerging neural head generative models Nguyen-Phuoc et al. (2020); Schwarz et al. (2020); Deng et al. (2022); Chan et al. (2021); Shi et al. (2021); Nguyen-Phuoc et al. (2019); Xue et al. (2022) adopt 3D-aware representations Mildenhall et al. (2021) which can be optimized by multi-view images through differentiable rendering. Though these implicit representations offer potential memory efficiency and structure complexity compared with traditional 3DMM-based representations Paysan et al. (2009); Blanz and Vetter (1999); Li et al. (2017). Query-based feature sampling and fully connected mapping slow down the convergence process. To maintain representation complexity while accelerate the optimization process, EG3D Chan et al. (2022) proposes tri-plane representation to explicitly store features on axis-aligned planes that are aggregated by a lightweight implicit feature decoder for efficient volume rendering. However, inherent coupling of features and mirroring issue are also brought with the efficiency. PanoHead An et al. (2023) propose to exploit extra in-the-wild data to supervise the back of head, thus can generate views in $360^o$ full head setting. Although it enriches the tri-plane's representational capacity through adding more parallel feature planes, PanoHead can not thoroughly solve the mirroring issue from representation level. SphereHead Li et al. (2024), through a shift in formulation from a Cartesian coordinate representation in cubic space to a spherical coordinate representation in spherical space, greatly eliminates the mirroring issue and avoids many artifacts. While SphereHead Li et al. (2024)

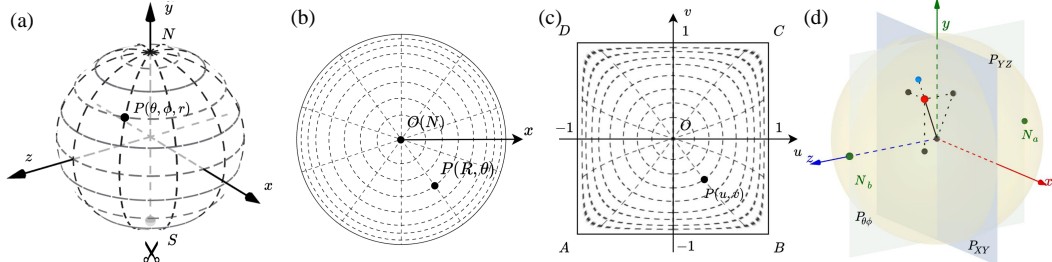

Figure 3: (a) The Lambert azimuthal equal-area projection (LAEA) opens the South Pole, and maps the sphere to (b) a flat circular plane, with the North Pole at its center. (c) Elliptical grid mapping transforms the circular plane into a square. Conversely, the square can be inversely mapped back to a sphere with near-equal-area properties. (d) In the hy-plane (2+2) representation, two spheres coincide, with their North Poles facing in opposite directions. Please refer to the supplementary videos for a more comprehensive understanding.

addresses many limitations of prior work, it fails to preserve the simplicity of representing symmetric objects and introduces discontinuities along the seam between the two poles. To this end, we propose a novel *hy-plane representation* that effectively represents both symmetric and asymmetric regions, eliminates the mirroring issue, and avoids representation discontinuity.

## 3    Method

### 3.1    Hy-Plane Representation

As illustrated in fig. 2, both tri-plane and spherical tri-plane have distinct strengths and limitations. The tri-plane representation benefits from uniform and dense spatial feature distribution, enabled by Cartesian coordinate projection. It efficiently renders high-resolution images from various angles and leverages symmetry effectively. However, it struggles with disentangling asymmetric features, leading to unwanted mirroring artifacts caused by feature entanglement. In contrast, the spherical tri-plane uses spherical coordinate projection to naturally distinguish directional features and learn anisotropic representations, avoiding feature entanglement. Yet, its non-uniform feature distribution reduces feature map utilization and complicates the rendering of high-resolution details.

Recognizing the complementary nature of these approaches, we propose *hy-plane*, a novel hybrid representation combining planar and spherical planes. hy-plane uses planar components to capture symmetric features and spherical components to model anisotropic features. This design retains the efficiency and uniformity of the tri-plane while eliminating feature entanglement and mirroring artifacts.

**Hy-Plane (3+1)** The basic version of our representation consists of three planar planes plus one spherical plane, referred to as hy-plane (3+1). The three planar planes are arranged mutually orthogonally, with the positive z-axis aligned toward the human face, the positive y-axis pointing to the top of the head, and the positive x-axis directed toward the left ear. Features are queried using Cartesian coordinate projection. The spherical plane adopts a spherical coordinate system, with the polar axis aligned with the head's top direction. Notably, instead of directly querying the feature map using $(\theta, \phi)$ coordinates, we employ a novel *near-equal-area warping* method to improve feature map utilization.

**Near-Equal-Area Warping** Directly querying the feature map using $(\theta, \phi)$ coordinates, as in spherical tri-plane, is straightforward but introduces significant side effects. Geometrically (fig. 2(d-f)), wrapping a square feature map $P_{\theta\phi}$ into a spherical plane causes numerical discontinuities at $\phi = -\pi$ and $\phi = \pi$, leading to artifacts in the seam region. Additionally, the edges $\theta = 0$ and $\theta = \pi$ contract into polar points, converting fluctuations along these lines into high-frequency noise around the poles. Furthermore, this warping is non-equal-area, unevenly distributing features from the square feature map onto the sphere. The equator region becomes feature-sparse, reducing expressive ability, while the poles become feature-dense, causing polar artifacts. Although SphereHead Li et al. (2024) addresses seam and polar artifacts by introducing an orthogonal dual-sphere setup, this approach doubles the number of feature planes but compromises the model's overall expressiveness, because

each sphere uses its feature-sparse equator region to cover the other's feature-dense polar regions, further limiting representation quality.

To address the challenging wrapping problem, we propose an elegant solution based on Lambert azimuthal equal-area projection (LAEA projection):

$$(R, \Theta) = \left( 2\cos\frac{1}{2}\phi, -\theta \right),$$ (1)

where $(\theta, \phi)$ denote colatitude and longitude in spherical coordinates on the spherical feature plane, and $(R, \Theta)$ denote radius and azimuth in polar coordinates on the flatten circle feature map. As shown in fig. 3 (a, b), LAEA projection unfolds the spherical surface from the South Pole and flattens it into a circular plane centered on the North Pole. This method ensures equal-area transformation by adaptively adjusting latitudinal line density along the radius, achieving uniform feature distribution during warping. Additionally, it consolidates the seam and two poles of the spherical coordinate system into a single point, making them easier to handle. We align this point with the downward direction of the 3D head, which remains invisible in the rendering.

Next, we use elliptical grid mapping to transform the circle into a square, as illustrated in fig. 3 (b, c), which can be formulated as follows:

$$(x, y) = (R\cos\Theta, R\sin\Theta),$$ (2)

$$\begin{cases} u = \dfrac{1}{2}\sqrt{2 + x^2 - y^2 + 2\sqrt{2}x} - \dfrac{1}{2}\sqrt{2 + x^2 - y^2 - 2\sqrt{2}x}, \\ v = \dfrac{1}{2}\sqrt{2 - x^2 + y^2 + 2\sqrt{2}y} - \dfrac{1}{2}\sqrt{2 - x^2 + y^2 - 2\sqrt{2}y}. \end{cases}$$ (3)

where $(x, y)$ are coordinates on the circle, and $(u, v)$ are coordinates on the square feature map. This near-equal-area mapping minimizes severe deformation, preserving feature quality. When querying a 3D point's feature on the spherical plane, we first convert its spherical coordinates $(\theta, \phi)$ to polar coordinates $(R, \Theta)$ on the wrapped circle using eq. (1). We then transform these polar coordinates into 2D Cartesian coordinates $(x, y)$ via eq. (2), and finally map them to the corresponding $(u, v)$ location on the square feature map using eq. (3). This approach maximizes the utilization of the feature map while effectively eliminating seam artifacts.

**Hy-Plane (2+2)** While in human head modeling, we can hide the final pole (the South Pole) by orienting it downward, this approach may not be applicable to other scenes or objects where no such unimportant direction exists for hiding. This limits its broader applicability. To address this, we introduce hy-plane (2+2), a variant consisting of two orthogonal planar planes and two spherical planes with opposing poles. When querying a 3D point's feature on the spherical planes, we compute features separately and combine them using a weighting function:

$$\begin{cases} w_a = (R_a^{\max} - R_a)^2, \\ w_b = (R_b^{\max} - R_b)^2, \\ f_{\text{sph}} = \dfrac{w_a f_a + w_b f_b}{w_a + w_b} \end{cases}$$ (4)

Here, $R_a$ and $R_b$ represent the radii of the 3D point projected onto the two wrapped circles. $R_a^{max}$ and $R_b^{max}$ denote the radii of the circles. The weights $w_a$ and $w_b$ are inversely proportional to these radii, peaking at the center ($R = 0$) and decreasing toward the edges ($R = R^{max}$). This design optimizes the use of the feature map's flat central region while minimizing the impact of the distorted edge areas, effectively resolving artifacts at the poles.

The reason for reducing one planar plane while adding a spherical plane is to be compatible with the unify-split strategy, as will be explained in section 3.2. Notably, as demonstrated in Wang et al. (2023a), two orthogonal planar planes can function nearly identically to three, since any two planes $(P_{XY}, P_{XZ}, P_{YZ})$ encompass all three coordinates $(x, y, z)$.

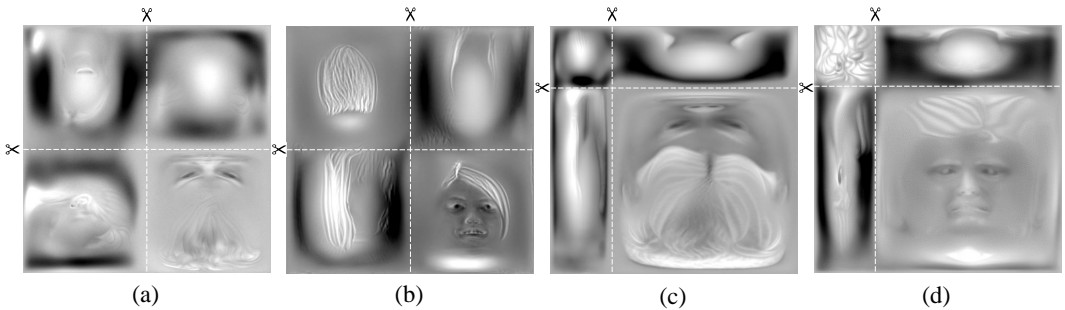

| (a) | (b) | (c) | (d) |

Figure 4: Unify-Split Strategy. (a) Hy-plane (3+1) with evenly splitting; (b) Hy-plane (2+2) with evenly splitting; (c) Hy-plane (3+1) with area-biased splitting; (d) Hy-plane (2+2) with area-biased splitting.

## 3.2 Unify-Split Strategy

**Feature Penetration across Channels** A key reason for the widespread adoption of tri-plane-like representations is their ability to represent 3D objects using 2D feature planes, whose data structure is similar to 2D images. This allows researchers to directly leverage existing 2D image generation architectures for 3D-aware object synthesis. However, reusing these models directly, without adapting to the inherent differences between 2D RGB images and 3D-aware tri-plane-like representations, leads to a critical oversight. In RGB images, the three channels represent different colors but share the same 2D spatial context. That is, the same uv position corresponds to the same spatial location, with only color variations across channels. This creates strong correlations during neural network training, enabling the network to first learn shared features layer by layer and then separate them into individual channels at the final output layer.

In contrast, in tri-plane-like representations, each plane encodes features from different spatial directions. Consequently, the same uv position on different planes corresponds to entirely distinct spatial meanings. Forcing convolutional networks to learn unrelated features at identical uv positions across planes increases learning complexity and causes feature entanglement between disparate spatial locations. This issue is particularly pronounced in 3D-aware GANs, where the model indirectly optimizes feature planes by learning from 2D images. The difficulty in disentangling information across feature planes leads to visible interference between output channels, resulting in unexpected artifacts in the generated images.

**Evenly Splitting** Based on the aforementioned observation, we adopt a simple yet effective *unify-split strategy* for synthesizing tri-plane-like representations. Instead of using separate channels for different feature planes, we allocate distinct regions on a large unified one-channel feature plane and then split it into parts corresponding to individual feature planes. This spatially disentangles features across planes in 2D space. fig. 4 (a, b) illustrates the splitting process for hy-plane (3+1) and hy-plane (2+2), where all four planes are evenly divided into two-by-two configurations.

**Area-Biased Splitting** Additionally, we can refine the 2D splitting scheme to enhance specific capabilities of the hy-plane. For instance, in full-head synthesis, the ability to model anisotropic features is crucial for generating high-resolution back-head details. As shown in fig. 4(c, d), for hy-plane (3+1), we increase the area of the spherical plane and elongate the feature maps $P_{XY}$ and $P_{YX}$ along different axes. This maximizes their expressive power when combined. For hy-plane (2+2), we enlarge one primary spherical plane while shrinking the other. The larger plane remains a full sphere, while the smaller one forms a spherical cap, covering the problematic polar region of the larger sphere.

## 3.3 HyPlaneHead

We integrate our hy-plane representation into HyPlaneHead, a 3D-aware GAN pipeline akin to Chan et al. (2022); An et al. (2023); Li et al. (2024). Given a sampled $z$ and conditioned camera parameter $c_{con}$, the generator $G$ produces a one-channel unified feature map, which is then split into individual feature planes of the hy-plane representation. Features are queried from each plane and volumetrically rendered using the viewing camera $c_{ren}$, enabling HyPlaneHead to generate a head image $I$ and

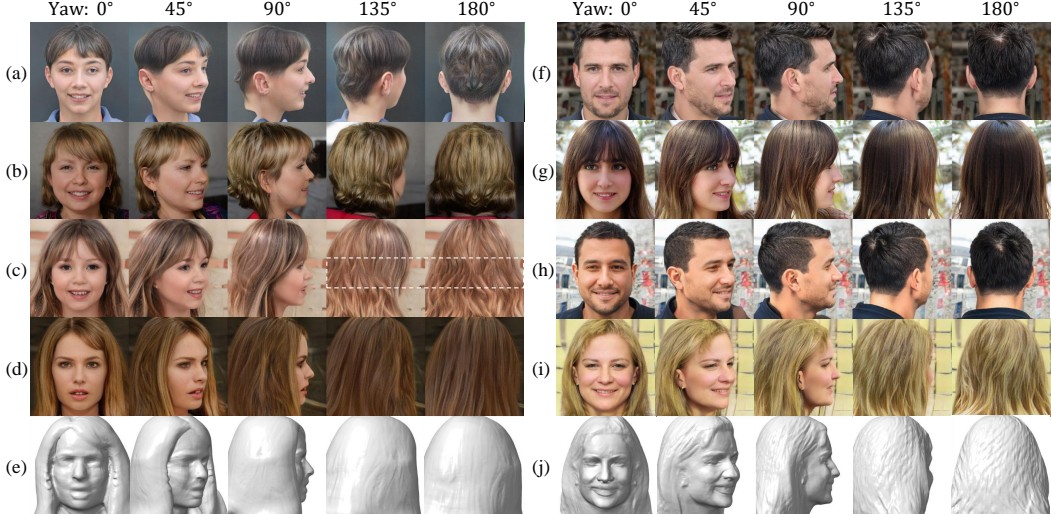

Figure 5: Qualitative comparison with state-of-the-art methods. (a) Tri-plane representation from Chan et al. (2022). (b) Tri-grid representation from An et al. (2023). (c) Single spherical tri-plane representation from Li et al. (2024), where the white dashed box highlights a discontinuity in the hair region. (d, e) Dual spherical tri-plane representation from Li et al. (2024). (f–j) Our proposed Hy-plane representation. A closer inspection is recommended for finer details, and we refer readers to the supplementary material for higher-resolution visualizations.

mask $I^m$. As in Chan et al. (2022); An et al. (2023); Li et al. (2024), the output passes through a super-resolution module to produce the high-resolution head image $I^+$. Following An et al. (2023); Li et al. (2024), we also introduce a background generator to allow $G$ to focus specifically on the head region. In addition to the conventional 3D-aware GAN losses used in Chan et al. (2022), we further employ a view-image consistency loss, as proposed in Li et al. (2024), to guide the discriminator to focus on the alignment between images and their corresponding viewpoints.

## 4 Experiments

In this section, we conduct comprehensive qualitative and quantitative experiments on full-head image synthesis to demonstrate that our hy-plane representation is well-suited for rendering from any viewpoint. Our comparative analysis includes tri-plane, tri-grid, spherical tri-plane from EG3D, PanoHead, SphereHead respectively, and various hy-plane variants and settings for ablation study, all trained on our dataset and pipeline. All experiments are trained on eight NVIDIA V100 GPUs with a batch size of 32. We follow PanoHead and SphereHead, using a training set that includes FFHQ Niemeyer and Geiger (2021), CelebA Liu et al. (2018), LPFF Wu et al. (2023b), WildHead Li et al. (2024), K-Hairstyle Kim et al. (2021), and a 6K in-house dataset of large-pose head images processed with SphereHead's toolbox. All training images are 512×512 in resolution and augmented with horizontal flips. The entire training process spans 25 million images.

### 4.1 Qualitative Comparison

We visualized synthesized samples and feature planes under varying configurations. For feature plane analysis, channel activations were averaged across feature-dimensions to visualize spatial texture patterns. As shown in fig. 1(a,b), tri-plane and spherical tri-plane models exhibit notable cross-plane interference: secondary feature maps show identical texture patterns from the dominating plane, alongside anomalous noise pattern. We suspect this phenomenon arises from inter-channel feature penetration, where competing planes disrupt each other's activations. Contrastingly, our unify-split strategy (fig. 1(c)) allows each plane to specialize in its directional features without cross-channel interference, thereby producing informative and clear feature maps. From fig. 1 (c) and fig. 4, our hy-plane representation effectively integrates planar and spherical planes, enabling seamless collaboration through a division of labor. The planar planes specialize in capturing symmetric

Table 1: Quantitative comparison on Full-head Image Synthesis task. We list FID, FID-random metrics with different representations. * denote that we output two spherical tri-planes simultaneously by a shared branch.

| No. | Representation | Unify-Split | Wrapping | FID | FID-random |
|---|---|---|---|---|---|
| 1 | Tri-plane (EG3D Chan et al. (2022)) | - | - | 9.22 | 11.23 |
| 2 | Tri-plane | evenly split | - | 8.86 | 11.52 |
| 3 | Spherical Tri-plane (SphereHead Li et al. (2024)) | - | - | 8.64 | 10.71 |
| 4 | Spherical Tri-plane | evenly split | - | 8.36 | 10.42 |
| 5 | Dual Spherical Tri-plane | - | - | 8.68 | 10.28 |
| 6 | Dual Spherical Tri-plane * | - | - | 11.9 | 13.54 |
| 7 | Tri-grid (PanoHead An et al. (2023)) | - | - | 8.77 | 10.66 |
| 8 | Tri-plane $512^2$ | - | - | 9.27 | 10.89 |
| 9 | Spherical Tri-plane $512^2$ | - | - | 8.82 | 10.47 |
| 10 | Tri-grid $512^2$ | - | - | 8.79 | 10.78 |
| 11 | Hy-plane (3+1) | - | - | 8.54 | 10.66 |
| 12 | Hy-plane (3+1) | evenly split | - | 8.31 | 10.18 |
| 13 | Hy-plane (3+1) | evenly split | yes | 8.18 | 9.96 |
| 14 | Hy-plane (3+1) | area-bias split | yes | **8.14** | 9.88 |
| 15 | Hy-plane (2+2) | evenly split | yes | 8.28 | 10.01 |
| 16 | Hy-plane (2+2) | area-bias split | yes | 8.17 | **9.84** |

features (e.g., $P_{YZ}$ learns left-right symmetric details such as side hair, ears, and shoulders), while the spherical plane excels at modeling anisotropic features (e.g., the frontal face and back-view hair).

fig. 5 compares our method with state-of-the-art full-head 3D-aware GANs. (a) The tri-plane representation from Chan et al. (2022), retrained on our dataset, generates mirrored faces at the back due to feature entanglement. (b) PanoHead An et al. (2023) uses a tri-grid structure but still exhibits excessive symmetry in hairstyles. (c) The single spherical tri-plane from SphereHead Li et al. (2024) produces full-head geometry but introduces seam and polar artifacts via $(\theta, \phi)$ warping. (d, e) Its dual spherical variant reduces artifacts but leads to over-smoothed textures and loss of detail. (f–j) Our HyPlaneHead combines planar and spherical representations, achieving high-quality synthesis with rich texture and geometric fidelity, setting a new benchmark for full-head 3D-aware GANs.

## 4.2 Quantitative Comparison and Ablation Study

To quantitatively evaluate the visual quality, fidelity, and diversity of the synthesized full-head images, we employed the Frechet Inception Distance (FID) metric Szegedy et al. (2016) on 50K real and synthetic samples. As noted in prior 3D-aware GANs based full-head image synthesis works An et al. (2023); Li et al. (2024), current 3D-aware GANs typically perform well under the conditioning camera pose during synthesis but degrade significantly at non-conditioned rendering angles. To rigorously assess performance under arbitrary viewing angles, which is especially critical for full-head image synthesis, we introduced a new evaluation metric, FID-random, which decouples the conditioning pose from the rendering pose. Specifically, during generation, we first randomly sample a camera parameter $c_{con}$ from the dataset's camera distribution to condition the tri-plane-like representation; subsequently, we render the head image using a different random camera parameter $c_{ren}$ (also sampled from the same distribution). The FID score is then calculated based on the images rendered under these random viewpoints, thereby providing an unbiased evaluation of the model's robustness and generalization across all possible angles.

Comparing table 1(1) with table 1(11) demonstrates the advantages of augmenting the tri-plane with a spherical plane, consistent with our earlier visualizations. The effectiveness of our unify-split strategy is evidenced by the general reduction in FID and FID-random scores across table 1(1,2,3,4,11,12). Notably, while applying the unify-split strategy to the tri-plane reduces FID, it increases FID-random. This occurs because the strategy eliminates inter-channel feature penetration, allowing each plane to fully express its directional features. However, since the tri-plane does not separate directional features, the enhanced plane expression exacerbates mirroring artifacts on the backside, thereby worsening FID-random. In contrast, both the spherical tri-plane and hy-plane benefit from the separation of directional features provided by the spherical plane, enabling them to leverage the improved expressiveness unlocked by the unify-split strategy, resulting in reductions in both FID and FID-random. table 1(3,4,5,6,9) reveal that directly outputting dual spherical tri-planes leads

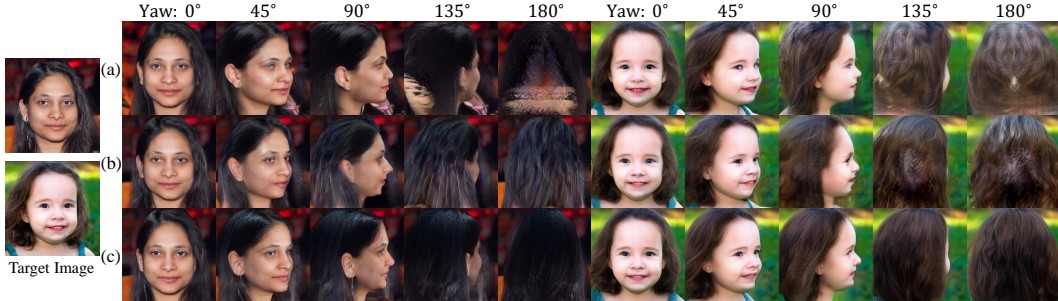

Figure 6: Single-view 3D-aware GAN inversion results. From top to bottom: (a) PanoHead An et al. (2023), (b) SphereHead Li et al. (2024), and (c) our HyPlaneHead.

to significant interference between the two dominant theta-phi planes, yielding the highest FID scores. SphereHead mitigates this issue by introducing two small convolution-based branches, albeit at the cost of increased parameters. However, adopting the unify-split strategy achieves superior results without additional parameters, as demonstrated by the single spherical tri-plane's performance. table 1(12,13) validate that wrapping improves performance by fully utilizing the square feature map. In table 1(14,16), we split a 512×512 feature map into four parts via area-bias splitting: 384×384, 384×128, 384×128, and 128×128, with the largest allocated to the spherical plane. Comparing these configurations with table 1(13,15) confirms the effectiveness of this partitioning scheme for full-head synthesis. Finally, we tested the tri-plane, tri-grid, and spherical tri-plane with a feature map size of 512×512. table 1(8,9,10) show minimal impact from increasing the feature map size, ruling out model parameter scaling as a significant factor influencing our experimental outcomes.

### 4.3 Single-view 3D-aware GAN Inversion

We compare our method with PanoHead and SphereHead on 3D full-head reconstruction from a single-view image using Pivotal Tuning Inversion (PTI) Roich et al. (2022). As shown in fig. 6, PanoHead consistently produces noticeable artifacts with strong left-right symmetry. SphereHead generates a plausible back-of-head region but yields blurry hair details, resulting in an overly coarse appearance. In contrast, our HyPlaneHead produces reasonable and high-quality renderings from all viewing angles.

## 5 Conclusion

In this paper, we conduct an in-depth analysis of the limitations inherent in tri-plane-like representations used in 3D-aware GANs, particularly focusing on mirroring artifacts, uneven warping from the square feature map to the spherical plane, and feature penetration across channels. Based on these insights, we propose the hybrid-plane (hy-plane) representation, which combines the strengths of planar and spherical planes while mitigating their respective weaknesses. Our technical contributions include a unified planar-spherical representation, near-equal-area warping for seamless and efficient square-to-sphere mapping, and a unify-split strategy to eliminate feature penetration. These innovations enable HyPlaneHead to achieve state-of-the-art performance in full-head image synthesis, significantly reducing artifacts and enhancing rendering quality.

## 6 Acknowledgements

The work was supported in part by NSFC with Grant No. 62293482, the Basic Research Project No. HZQB-KCZYZ-2021067 of Hetao Shenzhen-HK S&T Cooperation Zone, Guangdong Provincial Outstanding Youth Fund(No. 2023B1515020055), by Shenzhen Science and Technology Program No. JCYJ20220530143604010, NSFC No.62172348, the Shenzhen Outstanding Talents Training Fund 202002, the Guangdong Research Projects No. 2017ZT07X152 and No. 2019CX01X104, the Guangdong Provincial Key Laboratory of Future Networks of Intelligence (Grant No. 2022B1212010001), and the Shenzhen Key Laboratory of Big Data and Artificial Intelligence (Grant No. SYSPG20241211173853027).

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

# Supplementary Material

## A   Overview

This supplementary document provides additional materials to support the main paper *"HyPlaneHead: Rethinking Tri-plane-like Representations in Full-Head Image Synthesis"*. We present high-resolution qualitative comparisons that highlight the performance of our method against existing approaches in section B. A high-resolution qualitative comparison is provided in section C. We also provide a detailed explanation of the Near-Equal-Area Warping technique and our proposed Hy-Plane (2+2) representation, which are central to achieving view-consistent and artifact-free full-head image synthesis in section D. Comprehensive measurements of parameter count, training and inference speed, and VRAM usage across various methods and HyPlaneHead configurations are reported in section E. An in-depth discussion and comparison with related works, including OrthoPlanes, SYM3D, and other tri-plane-related algorithms, can be found in section F. Additional qualitative results across a broader set of examples are included to further demonstrate the effectiveness of our model in section G. We discuss the current limitations of our approach and potential directions for future work in section H. Finally, we include a section on Code of Ethics, where we address the ethical considerations and potential misuse of 3D head generation technologies in section I.

## B   High-Resolution Qualitative Comparison (Main Paper Fig. 5)

Due to the page limit of the paper, we were only able to include a low-resolution version of the qualitative comparison (Main Paper Fig. 5). To better demonstrate the advantages of our method in generating fine details, we provide a high-resolution version of Main Paper Fig. 5.

fig. 7 presents a high-resolution qualitative comparison with state-of-the-art methods. Each subfigure corresponds to the following representations: (a) Tri-plane representation from Chan et al. (2022). (b) Tri-grid representation from An et al. (2023). (c) Single spherical tri-plane representation from Li et al. (2024), where the white dashed box highlights a discontinuity in the hair region caused by seam artifacts. (d–e) Dual spherical tri-plane representation from Li et al. (2024). (f–j) Our proposed Hy-plane representation.

While both the tri-plane and tri-grid representations (a and b) yield rich details in front-views, they suffer from inherent symmetry artifacts due to their Cartesian coordinate projections. Specifically, (a) exhibits clear mirroring face artifacts on the back of the head, reflecting front-view facial attributes. Similarly, (b) shows excessive left-right symmetry in the rear view.

The single spherical tri-plane (c) addresses the symmetry issue by introducing a spherical projection. However, it introduces seam artifacts due to the discontinuity in the $(\theta, \phi)$ warping at the boundary of the spherical feature map (as shown in the white-dashed box, where the hair texture is misaligned).

To mitigate these seams, the dual spherical tri-plane approach (d–e) introduces an additional orthogonal spherical tri-plane. While this effectively eliminates seam artifacts, it comes at the cost of increased parameter numbers. Moreover, when merging the two spherical tri-planes, the regions with the lowest feature density—i.e., the equatorial areas—are used to cover the high-density polar regions of the other plane. This results in reduced expressiveness for fine details such as hair textures and shape contours.

In contrast, our method employs the Hy-plane representation, which leverages the dense and even spatial feature distribution of the tri-plane, as well as its efficient representation of symmetric regions, to ensure high-fidelity detail reconstruction. At the same time, a spherical tri-plane is utilized to provide anisotropic representation for asymmetric areas, effectively eliminating mirroring artifacts. Furthermore, we introduce a novel near-equal-area sphere-to-square warping strategy that avoids seam artifacts without compromising detail preservation.

## C   Details of Near-Equal-Area Warping

The Near-Equal-Area Warping method ensures that each region of the spherical input is mapped onto the planar representation with approximately equal surface area, add avoid excessive distortion

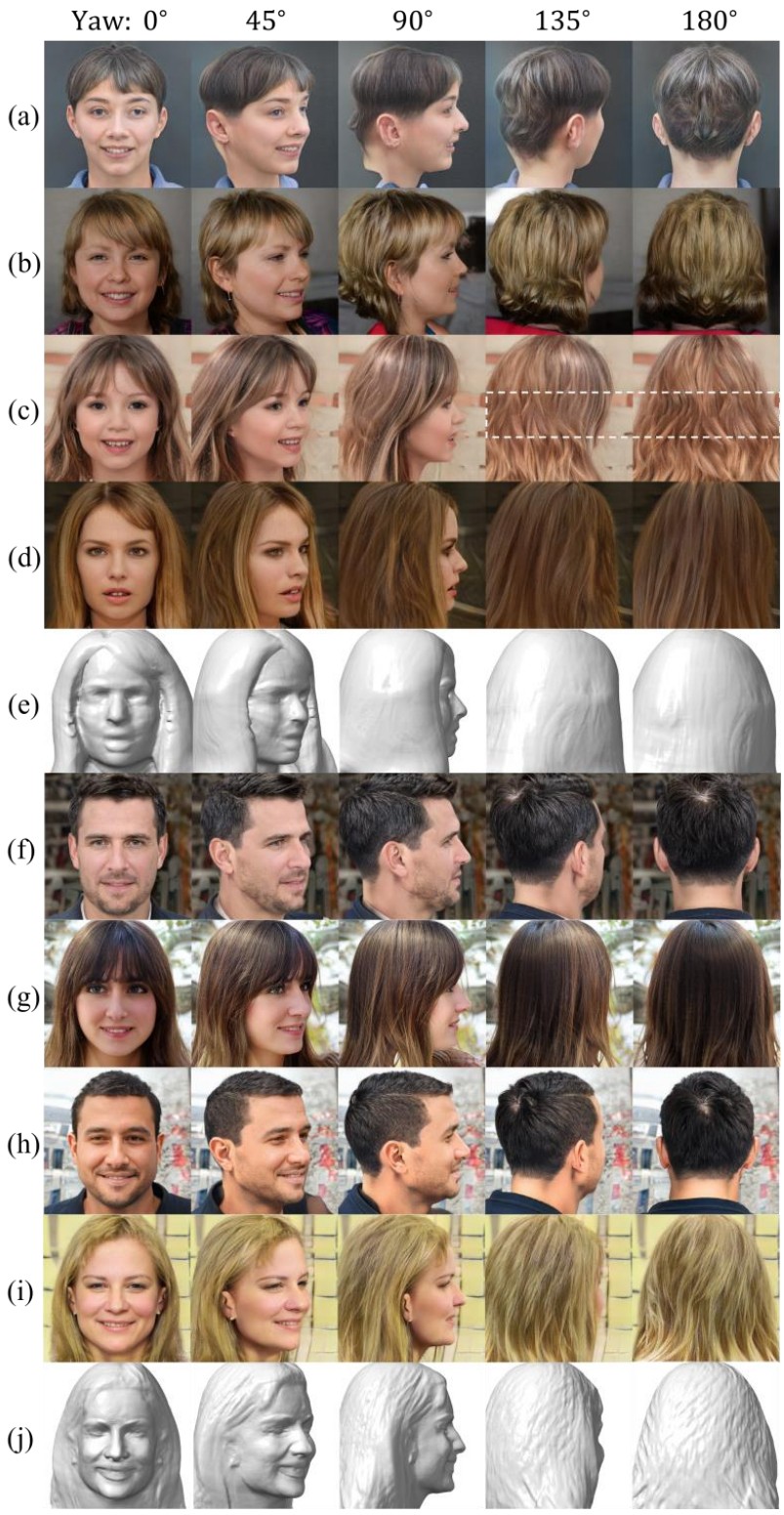

Figure 7: High-resolution qualitative comparison with state-of-the-art methods.

and eliminating seam artifacts. This warping strategy is implemented in two steps: first, we use the Lambert Azimuthal Equal-Area Projection (LAEA projection) to flatten the spherical surface into a

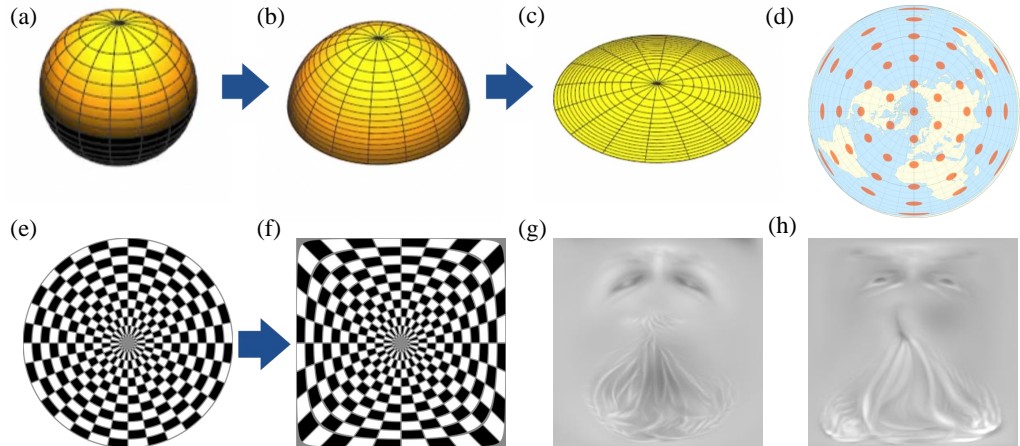

Figure 8: Illustrations of (a-d) the Lambert Azimuthal Equal-Area projection [*] and (e-h) Elliptical Grid Mapping [†].

circular domain while preserving area; second, we apply Elliptical Grid Mapping to transform the circle into a square domain, enabling efficient utilization of the square-shaped feature map.

To better understand our proposed Near-Equal-Area Warping, we provide additional details and illustrative diagrams in this supplementary material.

In the Lambert Azimuthal Equal-Area Projection, the south pole of the sphere is "opened" and then flattened into a circular domain centered at the north pole. During this unfolding process, the distances between latitude lines are adjusted such that the resulting circular projection maintains equal-area correspondence with the original spherical surface. A clearer understanding of this transformation can be gained from fig. 8. fig. 8(a–c) illustrates the dynamic process of the Lambert Azimuthal Equal-Area (LAEA) projection. fig. 8(d) illustrates the Lambert Azimuthal Equal-Area (LAEA) projection using a world map example, demonstrating that it preserves area. Each orange circle represents a region of equal size on the original spherical surface.

Subsequently, we employ Elliptical Grid Mapping to convert the circular domain into a near-equal-area square grid. Among various methods for transforming a circle into a square, we choose Elliptical Grid Mapping due to its following advantageous properties: 1. Approximate equal-area mapping: The variation in local area across the transformed plane is minimized. 2. Smooth central region and minimal distortion at the boundaries: This preserves important structural details, especially near edges. 3. Computationally simple and stable: It avoids division operations, which is crucial for maintaining gradient stability during training. An intuitive illustration of this mapping is provided in fig. 8. fig. 8(e,f) show the deformation of Elliptical Grid Mapping under black-and-white stripe patterns, indicating that most regions experience minimal area distortion. fig. 8(g,h) show the feature maps of the spherical plane before and after applying Elliptical Grid Mapping. Without this mapping, the model fails to effectively utilize the corner regions of the feature map. In contrast, with Elliptical Grid Mapping, most regions of the feature map are efficiently utilized.

For convenience, we have included the core implementation of the Near-Equal-Area Warping method in the supplementary material as `near_equal_area_warping.py`.

## D   Details of Hy-Plane (2+2)

Although the LAEA projection addresses seam artifacts, its implementation still requires "opening" the South Pole, which inherently leaves one remaining polar region. This region is more prone to high-frequency noise and distortion.

---

[*]`https://en.wikipedia.org/wiki/Lambert_azimuthal_equal-area_projection`
[†]`https://github.com/Kuuuube/Circular_Area/blob/main/wiki/mappings/elliptical_grid_mapping.md`

While in the Hy-Plane (3+1) formulation we can hide the problematic area by orienting it downward, this approach limits the generality of our representation, particularly for objects or scenes that require rendering from all directions. In such cases, relying on a single hidden region is insufficient, as any arbitrary viewpoint may expose the problematic area and lead to visible artifacts. To make the Hy-Plane representation more universally applicable, we propose Hy-Plane (2+2) to resolve this issue.

The Hy-Plane (2+2) representation combines two planar planes ($P_{XY}$ and $P_{YZ}$) and two spherical planes ($P_a$ and $P_b$). These two spherical planes overlap spatially but are oriented such that their respective South Poles face opposite directions. Specifically, as shown in main paper Fig. 3 (d), the North Pole of $P_a$ is oriented along the negative $z$-axis, while the North Pole of $P_b$ is oriented along the positive $z$-axis; consequently, their South Poles point in opposite directions. By assigning weights to each plane and summing them, the smooth North Polar regions of one spherical plane can effectively cover the problematic South Polar regions of the other, thereby eliminating the distortion-prone areas entirely.

# E  Parameter Count, Speed and VRAM Usage Comparison

To provide a comprehensive comparison of parameter count, training and inference speed, as well as VRAM usage across different methods and HyPlaneHead configurations, we conducted measurements for each experiment listed in Table 1 of the paper. The results are shown in table 2, where an asterisk (*) denotes that two spherical tri-planes are output simultaneously by a shared branch.

To provide a comprehensive and fair comparison, we clarify the definitions of the metrics used in our evaluation. Representation Parameters refer to the number of parameters in the feature maps of different tri-plane-like representations (e.g., tri-plane, spherical plane, hy-plane, etc.). Total Learnable Parameters denote the total number of trainable parameters in the entire model architecture, such as EG3D, PanoHead, SphereHead, and HyPlaneHead. Training Speed and Training VRAM Usage are measured on a single V100 GPU with a batch size of 2, representing the average time and memory consumption required to train 1,000 images. Similarly, Inference Speed and Inference VRAM Usage are evaluated under the same hardware setup but with a batch size of 1, by generating 100 images and computing the average time and memory cost per image.

From the statistics in the table, we can observe that different representations vary significantly in terms of feature plane parameter count. The tri-plane uses the fewest parameters ($3 \times 256 \times 256$ floating-point values), while the tri-grid with 512×512 resolution uses the most ($9 \times 512 \times 512$). Our proposed hy-plane uses $1 \times 512 \times 512$ floating-point values, which is only 1.33 times the number used by the tri-plane. However, the overall difference in total learnable parameters across models is relatively small. The minor variations are mainly due to differences in the final convolutional layer configuration of StyleGAN2, which depends on how each representation is generated.

Notably, the total learnable parameters for entries No. 3 and No. 5 are identical. This is because in experiment No. 3, we did not modify the model code at all, which means only the rendering pipeline was adjusted to use one spherical tri-plane. We reported the actual parameter count to remain consistent with our experimental setup.

In terms of training and inference speed, as well as VRAM usage, there are some differences among the experiments. Overall, however, the additional computational overhead introduced by our innovations is relatively small.

Replacing the tri-plane with hy-plane introduces:

- +5.5% training time,
- +3.8% inference time,
- +8.8% training VRAM,
- −0.8% inference VRAM.

Adding the evenly split strategy further increases the cost by:

- +8.4% training time,
- +6.9% inference time,

- +12.2% training VRAM,
- +16.6% inference VRAM.

When combining with the near-equal-area projection, the overhead becomes:

- +0.5% training time,
- +3.5% inference time,
- $-5\%$ training VRAM,
- $-5.7\%$ inference VRAM.

Please note that in some cases, VRAM usage may actually decrease, likely due to memory fragmentation causing inaccuracies in the `nvidia-smi` measurement.

In summary, while our method does introduce some computational and memory overhead, the increase is relatively modest and justifiable given the significant improvements in disentanglement and reconstruction quality.

# F  Comparison and Discussion with Related Works

In this section, we provide a detailed comparison of our hy-plane representation with two closely related recent works: OrthoPlanes He et al. (2023) and SYM3D Yang et al. (2024), and discussion on compatibility with several tri-plane-related algorithms.

## F.1  Comparison with OrthoPlanes

OrthoPlanes enhances the standard tri-plane representation by introducing multiple parallel planes along each Cartesian axis, effectively increasing the capacity of the feature field. This design is conceptually similar to the tri-grid formulation, where additional planar slices are used to capture finer geometric details.

However, this strategy does not address the underlying structural limitations of Cartesian-based projections. Specifically, OrthoPlanes still relies on axis-aligned planar projections in Cartesian Coordination, which inherently suffer from mirroring artifacts. As demonstrated in our main paper (fig. 1), such artifacts manifest as mirroring-face artifacts and unnatural left-right duplication. Our hy-plane representation mitigates this by integrating spherical projection planes that naturally align with the radial symmetry of human heads, thereby disentangling symmetric and asymmetric components more effectively.

Moreover, the increased number of planes in OrthoPlanes leads to higher storage overhead. For downstream applications such as 3D head reconstruction or model initialization (e.g., in Portrait3D Wu et al. (2024b), AnimPortrait3D Wu et al. (2025), or ID-Sculpt Hao et al. (2025)), each sample must store K times more feature maps (where K is the number of parallel planes per axis), significantly increasing memory and bandwidth requirements. In contrast, our hy-plane uses only four planes (three planar + one spherical) while achieving superior fidelity.

Finally, due to the increased number of planes, it becomes difficult to integrate these approaches with our unify-split strategy. As a result, inter-channel feature penetration remains an issue in these methods.

## F.2  Comparison with SYM3D

First, our goal differs from that of SYM3D. SYM3D enhances the symmetry of tri-plane representations through symmetric regularization, which is beneficial for generating fully symmetric artificial objects. In contrast, our hy-plane is designed to support a broader range of real-world scenarios where both symmetric and asymmetric structures coexist, such as in full-head portraits. Second, regarding feature penetration, SYM3D employs an attention-based scheme (View-wise Spatial Attention) to learn how to alleviate feature penetration across channels, whereas our hy-plane utilizes the unify-split strategy to geometrically and fundamentally prevent feature penetration at its source.

In fact, the correlation across $P_{XY}$, $P_{YZ}$ and $P_{XZ}$ planes discussed in SYM3D essentially corresponds to the inter-channel feature penetration problem we identify in our paper, though they observe

| No. | Representation | Unify-Split | Wrapping | Rep. Params | Total Params | Training Speed (sec/kimg) | Inference Speed (ms/image) | Training VRAM (MiB) | Infer. VRAM (MiB) |
|---|---|---|---|---|---|---|---|---|---|
| 1 | Tri-plane | - | - | 3×256×256 | 53,174,956 | 180.29 | 42.06 | 6436 | 1103 |
| 2 | Tri-plane | evenly split | - | 1×512×512 | 53,230,222 | 197.89 | 44.17 | 7670 | 1149 |
| 3 | Spherical Tri-plane | - | - | 3×256×256 | 54,713,868 | 222.84 | 54.99 | 8048 | 1481 |
| 4 | Spherical Tri-plane | evenly split | - | 1×512×512 | 53,234,479 | 198.57 | 48.19 | 7692 | 1097 |
| 5 | Dual Spherical Tri-plane | - | - | 6×256×256 | 54,713,868 | 245.79 | 54.87 | 9296 | 1481 |
| 6 | Dual Spherical Tri-plane * | - | - | 6×256×256 | 53,462,509 | 196.88 | 50.42 | 6810 | 1235 |
| 7 | Tri-grid | - | - | 9×256×256 | 53,741,548 | 198.59 | 45.19 | 6836 | 1245 |
| 8 | Tri-plane $512^2$ | - | - | 3×512×512 | 53,423,246 | 181.00 | 47.77 | 6450 | 1333 |
| 9 | Spherical Tri-plane $512^2$ | - | - | 6×512×512 | 54,962,158 | 182.28 | 59.91 | 6744 | 1103 |
| 10 | Tri-grid $512^2$ | - | - | 9×512×512 | 54,002,318 | 266.13 | 58.79 | 8642 | 2463 |
| 11 | Hy-plane (3+1) | - | - | 4×256×256 | 53,269,388 | 190.40 | 43.64 | 6966 | 1095 |
| 12 | Hy-plane (3+1) | evenly split | - | 1×512×512 | 53,230,222 | 206.54 | 46.27 | 7818 | 1277 |
| 13 | Hy-plane (3+1) | evenly split | yes | 1×512×512 | 53,230,222 | 207.21 | 47.89 | 7432 | 1205 |
| 14 | Hy-plane (3+1) | area-bias split | yes | 1×512×512 | 53,230,222 | 226.31 | 49.61 | 7558 | 1321 |
| 15 | Hy-plane (2+2) | evenly split | yes | 1×512×512 | 53,230,222 | 212.27 | 49.81 | 7780 | 1215 |
| 16 | Hy-plane (2+2) | area-bias split | yes | 1×512×512 | 53,230,222 | 219.73 | 51.74 | 8012 | 1255 |

Table 2: Comparison of different 3D representation configurations in terms of parameter count, training/inference speed, and VRAM usage.

it from a different angle. We detect this issue through visual inspection of feature maps (as shown in fig. 1(a,b)), while SYM3D quantifies it using correlation metrics.

Inter-channel feature penetration causes similar values at the same UV positions across different feature maps, resulting in visually similar patterns, exactly what can be seen in our fig. 1(a,b). This phenomenon is also visible in the top-right panel of fig. 7 in SYM3D, where GET3D's feature maps exhibit repetitive vertical lines at the same spatial locations when zoomed in. Numerically, this leads to high correlations between different feature maps.

The difference lies in terminology: SYM3D does not explicitly refer to this as inter-channel feature penetration, but rather describes it as correlation. SYM3D attributes this issue to the use of single-view images during training, as opposed to multi-view data. While this is certainly true—more views provide richer geometric cues and reduce ambiguity—we also discuss this point in our paper. From a more fundamental perspective, however, the issue arises due to the structural nature of convolutional networks, where different channels are prone to interfere with each other, especially in the absence of direct supervision. Our solution addresses this root cause directly by modifying the architecture via the unify-split strategy, which effectively eliminates inter-channel feature penetration at its source.

In contrast, SYM3D uses view-wise spatial attention to alleviate the issue. As shown in the bottom-right panel of fig. 7 in SYM3D, this approach does reduce correlation to some extent. However, as illustrated in the middle-bottom panel, strong correlations still exist between certain planes, particularly $P_{YZ}$ and $P_{XZ}$. We have also computed and visualized similarity matrices similar to those in SYM3D's fig. 7, and our results show significantly lower feature correlations across different planes.

Moreover, in terms of implementation complexity, our unify-split strategy is simpler and more intuitive compared to SYM3D's view-wise attention mechanism, while achieving a more complete resolution of the problem.

### F.3 Discussion on Compatibility with Tri-plane-related Algorithms

Dual Encoder Bilecen et al. (2024) introduces a dual-encoder GAN inversion approach for single-view 3D full-head reconstruction. It uses one encoder for the visible front region and another for the occluded back region, addressing issues such as mirroring artifacts in PanoHead's W space. In contrast, our HyPlaneHead improves the underlying representation structure to reduce such artifacts and enhance W space quality. As a result, standard inversion methods like PTI—a general-purpose technique for common GANs—already yield better performance than previous approaches. We believe that combining our method with specialized inversion strategies like Dual Encoder, which are specifically tailored for full-head generation, could further improve results.

Tri2-plane Song et al. (2024) introduces a cascaded tri-plane representation across multiple scales of facial features, similar to a feature pyramid. This hierarchical design enables the model to generate both global structure and fine-grained details, resulting in richer and more detailed head reconstructions. Although it is still based on the tri-plane formulation and thus remains susceptible to mirroring artifacts, we believe that the multi-scale feature pyramid concept could be beneficially integrated with our hy-plane representation in future work.

Our method is also compatible with recent local editing approaches, such as Bilecen et al. (2025), which has been successfully applied to both tri-plane (EG3D) and tri-grid (PanoHead) representations. Given that our hy-plane shares a similar structure, we believe the method should be directly applicable to our representation as well.

Inspired by recent 3D avatar approaches Qian et al. (2024); Zhang et al. (2025); Qiu et al. (2025b); Kirschstein et al. (2025); Chu and Harada (2024); Qiu et al. (2025a); Kirschstein et al. (2024) based on 3D Gaussian Splatting (3DGS) Kerbl et al. (2023), we believe our hy-plane representation can be effectively integrated with 3DGS to further enhance rendering quality. We plan to explore this promising direction in future work.

Additionally, we believe that replacing the standard tri-plane representation in methods like LRM Hong et al. (2023) and InstantMesh Xu et al. (2024) with our hy-plane has the potential to improve their generation quality. On one hand, our unify-split strategy eliminates inter-plane feature penetration, allowing each plane to express its features more clearly and effectively. On the other hand, by incorporating a spherical plane, we enhance the model's ability to represent asymmetric regions, such

as facial details and hair on the back of the head. Therefore, we expect that integrating hy-plane into these models would lead to more accurate and artifact-free 3D reconstructions.

## G   More Qualitative Comparison

Figure 9 shows additional qualitative comparisons between our method and existing approaches on a larger set of examples. On the left side, samples (1–2) are generated by the official EG3D model, which is trained exclusively on the FFHQ dataset that lacks large-pose and back-view head images. As a result, these samples exhibit severe mirroring artifacts in their back views. Samples (3–4) use the tri-plane representation trained within our pipeline and data; while hair appears in the rear region, the front-facing facial attributes still dominate, indicating incomplete adaptation to non-frontal views. Samples (5–7) come from the official PanoHead model, which can produce detailed facial and hair textures but suffers from strong left-right symmetry and visible artifacts in the back view. Similarly, samples (8–10), using the tri-grid representation trained with our setup, also exhibit comparable symmetry and artifact issues. These problems—mirroring artifacts and severe left-right symmetry—are primarily caused by the Cartesian projection used in both tri-plane and tri-grid representations. Samples (11–13) are from the official SphereHead model, which addresses the mirroring issue through its spherical tri-plane design but results in more blurred outputs with less detailed facial and hair textures. The same trend is observed in samples (14–16), where a spherical tri-plane model is trained using our pipeline and data. In contrast, on the right side, samples (17–32) are generated by our HyPlaneHead model. By leveraging the novel hy-plane representation, our method not only eliminates mirroring artifacts but also achieves high-quality, consistent rendering from arbitrary view angles.

## H   Limitations

Despite the promising results of our method, there are still several limitations that warrant further investigation. First, similar to previous works such as EG3D, PanoHead, and SphereHead, our model exhibits minor visual flickering or instability in fine details when rendering from gradually changing viewpoints. We believe this is partly due to the current GAN backbone's limited capacity for high-fidelity view-consistent generation, and we plan to address this by adopting a more powerful generator architecture in future work.

Second, our method, like existing approaches, struggles with generating highly complex hairstyles such as ponytails, braids, or other structured hair arrangements. This limitation likely stems from insufficient training data covering such styles. We regard this as an important direction for future research and intend to expand our training dataset to include a broader variety of hairstyles and appearances.

## I   Code of Ethics

Our work presents a method for learning generalizable 3D full-head modeling from monocular images, which has potential applications in virtual avatars, digital content creation, and immersive experiences. However, such technology also raises ethical concerns, particularly regarding privacy and the potential for misuse, such as identity deception or unauthorized generation of realistic 3D head models. We are aware of these risks and emphasize the importance of responsible deployment, transparency, and user consent in any real-world application of this technology.

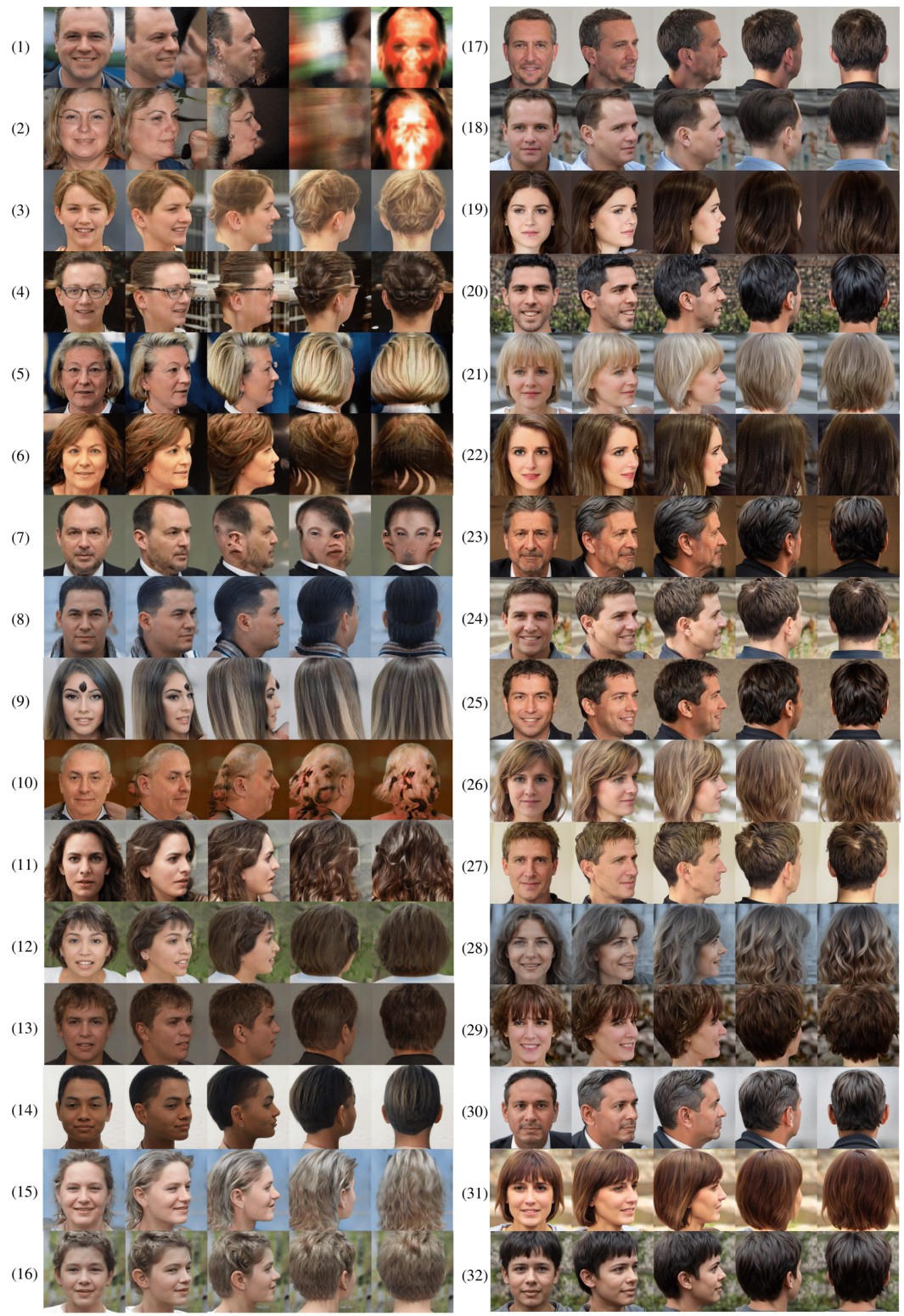

Figure 9: Additional qualitative results.

