# OpenReview forum: "HyPlaneHead: Rethinking Tri-plane-like Representations in Full-Head Image Synthesis"
_NeurIPS.cc/2025/Conference — NeurIPS 2025 poster_

### Official Review · Reviewer_QRHG · 2025-06-27

**Clarity:** 4
**Significance:** 3
**Originality:** 3
**Rating:** 5
**Confidence:** 5

**Summary:**

This paper addresses the challenging task of generating high-fidelity 3D head avatars. Building upon the EG3D framework [1], the authors introduce a novel hybrid triplane representation that effectively combines spherical and planar planes. To enhance feature learning, they propose two key contributions: Unify-Split Strategy, preventing feature disentanglement across triplane channels by unifying and then splitting feature maps and improved warping Formulation, a refined warping method for better spherical plane representation. Their approach is highly flexible, allowing adjustments in the number of planes and their configurations to suit different tasks. Extensive experiments demonstrate superior performance over existing methods, including recent state-of-the-art techniques like PanoHead [2] and SphereHead [3], both quantitatively and qualitatively.

**Questions:**

1. Could the merged triplane structure with single channels and splitting approach support triplane-localization as detailed in [4] for fine grained editing?
2. Could the authors show what each of the planes learn/represent in the output domain by just sampling from planar planes or single spherical planes, if possible.
3. Methods like LRM [5] and InstantMesh [6] also utilize triplane structure. In your opinion, how would your method improve the generation quality of those models?
4. Could the authors provide some examples of single view GAN inversion using your model’s triplane representation as in [2]?
5. The 2+2 approach should be supported further, since there are no visual samples to compare with 3+1. Could the authors provide an accompanying qualitative comparison to Table 1, where visual outputs of No 11-16 are compared? With this new figure, it would be also interesting to see how mapping network conditioning camera parameter affects the rendering results, with different camera parameters (FID vs. FID-random).

I am willing to increase my score a point once the questions and weaknesses have been adequately addressed.

[1] Chan et al., EG3D: Efficient Geometry-aware 3D Generative Adversarial Networks, CVPR 2022

[2] An et al., PanoHead: Geometry-Aware 3D Full-Head Synthesis in 360°, CVPR2023

[3] Li et al., SphereHead: Stable 3D Full-head Synthesis with Spherical Tri-plane Representation, ECCV2024

[4] Bilecen et al., Reference-Based 3D-Aware Image Editing with Triplanes, CVPR2025

[5] Hong et al., LRM: Large Reconstruction Model for Single Image to 3D, ICLR 2024

[6] Xu et al., InstantMesh: Efficient 3D Mesh Generation from a Single Image with Sparse-view Large Reconstruction Models, 2024

[7] Bilecen et al., Dual Encoder GAN Inversion for High-Fidelity 3D Head Reconstruction from Single Images. NeurIPS 2024

[8] Tri2-plane: Thinking Head Avatar via Feature Pyramid, 2024

**Ethical Concerns:**

["NO or VERY MINOR ethics concerns only"]

**Final Justification:**

I find the rebuttal adequate, hence I will increase my final score. However, I expect to see the results of Q4 and Q5 to be in the final paper & supplementary material.

**Limitations:**

The limitations are adequately addressed.

**Paper Formatting Concerns:**

No concerns.

**Quality:**

4

**Strengths And Weaknesses:**

Strengths:
1. The paper is well-written with great explanations and visuals in provided in the supplementary material.
2. The paper nicely identifies the feature penetration issue present in the multi-channel triplane representation and proposes a method called Unify-Split Strategy to mitigate the issue in a novel and intuitive way.
3. The flexibility of splitting is well-presented. The connection between (2+2) representation of spherical planes and ability to make one region smaller while keeping other larger to “cover” the problematic areas is a good idea.
4. The method is generalizable to different needs. As the authors state, while using 3+1 combination for head domain one could hide the disappearing pole at the bottom of the head but when it comes to general domain they need two spherical planes with their south poles facing opposite ways. The fact that the method can support different needs with simple modifications shows the generalizability of the method.
5. Their approach solves the low quality 3D structure problem present in SphereHead due to merging of spherical planes in an elegant way by combining planar planes and spherical planes.
6. Using LAEA (Lambert Azimuthal Equal-Area) Projection is a good idea as the authors explain the shortcomings of using spherical coordinates such as uneven feature distribution and numerical discontinuities.

Weaknesses:
1. Even though it is mentioned in the checklist, the paper does not detail the VRAM usage and a comparison for inference durations for different models. This is needed to measure how much overhead is added by the introduction of hybrid representation.
2. The authors only use FID metric to measure quantitive performance. Supporting FID with IS (Inception Score) as in [2] could better support the findings.
3. The authors do not include results for diverse ethnicities and skin colors. Addition of such results are required to show the model’s ability to generalize.
4. For the sake of completeness, it would be good to add some other related works based on triplanes [7,8] to the discussion.

---

> ### Author Rebuttal · Authors · 2025-07-28
>
> Thanks for your insightful review! Here are our responses to your concerns. We will include the following content in the final version of the paper and the supplementary material.
>
> **W1: Inference Speed and VRAM Using**
>
> *Due to space limitations, please refer to our reply to Reviewer Xp1z, Q1 & Q2*
>
> **W2: Additional Quantitative Metrics**
>
> Following PanoHead, we compute the Inception Score (IS) for our HyPlaneHead model. The IS results of Table 1 (from No. 1 to No. 16) are as follows:
> 3.72, 3.86, 3.99, 4.06, 3.96, 3.61, 4.01,
> 3.76, 4.03, 3.97,
> 4.10, 4.17, 4.22, 4.18, 4.21, 4.21
>
> **W3: Diversity in Qualitative Results**
>
> Similar to previous works (e.g., EG3D, PanoHead, SphereHead), our training data includes widely used portrait datasets such as FFHQ, CelebA. These datasets are typically collected from public sources, where Caucasian individuals constitute the majority, while East Asian and South Asian individuals are present in smaller proportions. Other ethnic groups are underrepresented. As a result, the data distribution directly influences the randomly sampling results of the model. Therefore our method tends to generate predominantly Caucasian faces, with occasional East or South Asian characteristics, as seen in Supplementary Figure 3 (images 30 and 32).
>
> However, this does not necessarily mean that our model is unable to handle underrepresented ethnicities or skin tones. Instead, it reflects the fact that the sampling distribution aligns with the distribution of the training data. By increasing the representation of diverse ethnicities and skin tones in the training set, we expect the model to generate a broader range of identities accordingly.
>
> Moreover, when using GAN inversion techniques (e.g., PTI) on images from underrepresented groups, such as those of African descent, we consistently observe reasonable reconstructions of full-head 3D models. This is because our model has learned a rich and accurate distribution of head shapes and appearances through exposure to large-scale data, which provides a certain level of generalization. For out-of-distribution samples, targeted adaptation via inversion can further yield reasonable results.
>
> Since we are unable to include these results here via figures or external links, we will add them to the final version of the manuscript.
>
> **W4: Discussion with Related Work**
>
> Dual Encoder [7] introduces a dual-encoder GAN inversion approach for single-view 3D full-head reconstruction. It uses one encoder for the visible front and another for the occluded back region, addressing issues like mirroring artifacts in PanoHead’s W space.
>
> In contrast, our HyPlaneHead improves the representation structure to reduce such artifacts and enhance W space quality. As a result, standard inversion methods like PTI, which is a general-purpose technique for common GANs, already yield better performance than previous approaches. We believe that combining our method with specialized inversion strategies like Dual Encoder, which are specifically tailored for full-head generation, could further improve results. We will explore this possibility in the final version of the paper.
>
> Tri2-plane [8] introduces a cascaded triplane representation across multiple scales of facial features, similar to a feature pyramid. This hierarchical design allows the model to generate both global and fine-grained details, leading to richer and more detailed head reconstructions. We believe that the idea of multi-scale feature pyramids could be beneficial to integrate with our hy-plane representation in future work.
>
> **Q1: Compatibility with Triplane-Localization for Editing**
>
> We have carefully read the paper, and find the proposed triplane-localization method has been successfully applied to both tri-plane (EG3D) and tri-grid (PanoHead) representations.
>
> Given that our hy-plane shares a similar structure, we believe the method should be directly applicable to our representation as well. Moreover, we do not expect the combination of planar and spherical planes or the use of the unify-split strategy to interfere with the application of this localization approach.
>
> **Q2: Interpretability of Individual Planes**
>
> Since we are unable to include figures, we provide a qualitative explanation instead.
>
> Firstly, when rendering using only a single planar or spherical plane, the resulting output is barely informative. This is because, for a planar plane, all rays passing through the same point on that plane share the same feature value, leading to visual artifacts such as parallel lines that reflect the shape of the feature map itself. Similarly, when visualizing only the spherical plane, the output consists of many radial lines extending from the center of the sphere, which do not convey meaningful 3D structure.
>
> Instead, we can infer the contribution of each plane by removing one feature map at a time and observing the impact on the final rendering.
>
> When the spherical plane is removed among the four feature planes, we observe significant missing regions in the facial area and back-of-the-head hair in the rendered result. This indicates that the spherical plane has learned to represent asymmetric features such as the front face and rear hair. However, structures on both sides, such as the shoulders and ears, remain mostly present, though with some degree of distortion.
>
> When any one of the three planar planes (e.g., Pxy, Pyz, Pxz) is removed, significant distortions appear in the reconstructed head shape. This is because, in addition to carrying meaningful feature information, these planes also serve a role similar to "positional encodings" during training. If one of them is absent during inference, the positional representation becomes inconsistent with that used during training, resulting in structural degradation.
>
> Among the three, the Pyz plane contributes the most to 3D structure, as its removal leads to a noticeable loss of symmetric features such as ears and shoulders. Removing the other two planar planes also introduces distortion, though to a lesser extent.
>
> **Q3: Discussion with Other Triplane-Based Methods (LRM, InstantMesh)**
>
> We believe that replacing the standard triplane representation in methods like LRM [5] and InstantMesh [6] with our hy-plane has the potential to improve their generation quality.
>
> On one hand, our unify-split strategy eliminates inter-plane feature penetration, allowing each plane to express its features more clearly and effectively. On the other hand, by incorporating a spherical plane, we enhance the model’s ability to represent asymmetric regions, such as facial details and hair on the back of the head.
>
> Therefore, we expect that integrating hy-plane into these models would lead to more accurate and artifact-free 3D reconstructions.
> Due to time constraints during the rebuttal phase and the lack of prepared data for such integration experiments, we have not yet conducted these evaluations. We will explore this direction experimentally in future work and include the results in the final version of the paper.
>
> **Q4: Single-View GAN Inversion Examples**
>
> *Due to space limitations, please refer to our reply to Reviewer Xp1z, Q3.*
>
> **Q5: Qualitative Comparison between 2+2 and 3+1 Configurations & FID vs. FID-random**
>
> Since we are unable to include additional figures in this response, we would like to describe our findings qualitatively here and will incorporate detailed visual comparisons into the final version of the manuscript.
>
> No. 11, which does not use the unify-split strategy, still suffers from inter-channel feature penetration. This is visible as irregular horizontal patterns in the hair or color artifacts on clothing, which is actually because of features that leak across different channels. Nevertheless, the overall quality is already better than the baseline methods.
>
> No. 11 and 12 also lack the near-equal-area warping, resulting in slightly blurred fine details, especially in the texture of the back-of-the-head hair.
>
> The results of No. 13–16 show very small differences, which can only be observed through careful and extensive comparison. The main distinction lies in the orientation of the spherical plane. In No. 15 and 16, one of the spherical planes has its north pole aligned with the front of the face. Since the LAEA projection minimizes distortion around the pole, these configurations capture more facial details, leading to a slightly sharper facial appearance. Meanwhile, No. 14 and 16, which use an area-biased split, enhance the expressive power of the spherical plane, allowing for better reconstruction of asymmetric features, which is consistent with the quantitative results in Table 1.
>
> Regarding FID vs. FID-random, as discussed in the paper, we observe that 3D-aware GANs tend to generate higher-quality results in regions aligned with the conditional camera view. For example, when the camera is conditioned to the left side of the head, the left side is well-generated, while the right side tends to be blurry or distorted. This is why we introduced the FID-random metric to evaluate how well the model generalizes across arbitrary viewpoints.
>
> Across all 3D-aware GANs, it is common for FID-random to be higher than FID, and they reflect the same trend. However, we want to highlight an interesting anomaly in No. 2: although adding the unify-split strategy to tri-plane reduces FID, it actually increases FID-random. This occurs because the strategy eliminates inter-channel feature penetration and allows each plane to express its directional features more fully. However, since the tri-plane does not explicitly separate directional information, this enhanced expression leads to stronger mirroring artifacts on the backside of the head, thereby worsening the FID-random score.

---

### Official Review · Reviewer_abHE · 2025-07-02

**Clarity:** 3
**Significance:** 3
**Originality:** 3
**Rating:** 4
**Confidence:** 4

**Summary:**

This paper analyses the limitations inherent in triplane-like representations used in 3D-GAN, and introduces the hy-plane representation, combining the strengths of both planar and spherical planes. To maximise its representation ability, a series of improvements are proposed. Experiments clearly demonstrate the advantages of the proposed representation.

**Questions:**

* Please clarify what form of disentanglement the proposed hybrid representation intends to achieve. It would help to include consistent visualizations or quantitative metrics (e.g., view separation score, plane correlation matrix) that support this claim.

* How does your method compare to recent methods that also address triplane entanglement, such as OrthoPlanes [1] or SYM3D [2], a discussion is encouraged.

* In your qualitative results, each method seems to use different conditions or reference views. How does your model perform in single-view settings under the same condition as other methods?

**Ethical Concerns:**

["NO or VERY MINOR ethics concerns only"]

**Final Justification:**

The rebuttal addressed most of my concerns.

The authors should include the additional results in the rebuttal, especially W4, and relation to the previous works (W1-2) in the revised paper.

**Limitations:**

Yes

**Quality:**

2

**Strengths And Weaknesses:**

Strengths:

* The motivation of this work is clear, and the proposed method appears practical and feasible.

* The mixing algorithm for planar and spherical could generalize beyond the current application to other triplane-based 3D representations.

* The ablation studies have shown how each component contributes.

Weakness:

* The disentanglement issue in triplane representations has been previously studied in works such as [1] and [2]. For example, [1] introduces additional depth dimensions within each triplane slice, and [2] applies symmetric regularization to improve cross-plane consistency.

* In [2], the core problem is identified as the correlation across xy, yz, and xz planes, primarily caused by single-view synthesis. In contrast, multi-view synthesis tends to better capture front, bottom, and side views. In Figure 4 of the paper, it remains unclear how the proposed method effectively addresses this disentanglement.

* The visualization does not clearly show distinct view-specific attributes. Clarifying the type of triplane disentanglement targeted by this work would strengthen the contribution.

[1]OrthoPlanes: A Novel Representation for Better 3D-Awareness of GANs
[2]SYM3D: Learning Symmetric Triplanes for Better 3D-Awareness of GANs

* The issue of feature penetration across channels is discussed, but no quantitative metric is provided to evaluate this property. The paper would benefit from a clearly defined metric or analysis to assess this claim.

* For the qualitative comparisons, the input or reference views vary across different methods, making direct visual comparison difficult. Including a consistent single-view reconstruction task, with matched inputs across all methods, would provide a fairer evaluation of visual fidelity and structural consistency.

---

> ### Author Rebuttal · Authors · 2025-07-28
>
> Thanks for your insightful review! Here are our responses to your concerns.
>
> **W1 & Q2: Discussion and Comparison with Prior Works**
>
> Thank you for pointing out these related works. We agree that discussing them is essential to the completeness of our paper, and we will incorporate the following analysis into the final version.
>
> Comparison with OrthoPlanes:
>
> The main difference between our method and OrthoPlanes lies in their design choices. OrthoPlanes introduces additional parallel planes within tri-plane to enhance expressiveness, which is similar to the tri-grid. In contrast, our method achieves disentanglement and effective learning of symmetric and asymmetric features by combining planar planes with spherical planes.
>
> As a result, OrthoPlanes still suffers from the following limitations, which are effectively addressed in our approach:
> 1. Incomplete resolution of mirroring artifacts: The root cause of mirroring artifacts lies in the use of Cartesian coordinate projection when querying features. As a result, both OrthoPlanes and tri-grid still suffer from mirroring-face artifacts and excessive left-right symmetry in certain regions.
> 2. Increased storage requirements: When saving a sample (e.g., for downstream tasks such as 3D model reconstruction or initialization for methods like Portrait3D [1], AnimPortrait3D [2], and ID-Sculpt [3]), they must store K times more feature planes (where K is the number of added parallel planes) leading to a significant increase in memory usage.
> 3. Incompatibility with unify-split strategy: Due to the increased number of planes, it becomes difficult to integrate these approaches with our unify-split strategy. As a result, inter-channel feature penetration remains an issue in these methods.
>
> Comparison with SYM3D:
>
> 1. Goal Difference
>     - SYM3D enhances the symmetry of tri-plane representations through symmetric regularization, which is beneficial for generating *fully symmetric artificial objects*.
>     - Our hy-plane is designed to support a broader range of real-world scenarios where *both symmetric and asymmetric structures coexist*, such as in full-head portraits.
>
>     Nevertheless, we believe that the symmetric regularization used in SYM3D could be beneficial if selectively applied to the planar planes in our method, especially the Pyz plane, which is responsible for learning symmetric facial features such as ears. We will explore this possibility in future work.
>
> 2. Different Strategies to Address Feature Penetration
>     - SYM3D employs an attention-based scheme (View-wise Spatial Attention) to *learn how to alleviate* feature penetration across channels.
>     - Our hy-plane, utilizes the unify-split strategy to *geometrically and fundamentally prevent* feature penetration at its source.
>
>
> [1] Portrait3D: Text-Guided High-Quality 3D Portrait Generation Using Pyramid Representation and GANs Prior
>
> [2] Text-based Animatable 3D Avatars with Morphable Model Alignment
>
> [3] ID-Sculpt: ID-aware 3D Head Generation from Single In-the-wild Portrait Image
>
> **W2: Discussion and Comparison with SYM3D**
>
> Thank you for introducing this related work. This is a very insightful and meaningful comparison, as it supports our findings from a different perspective.
>
> In fact, the correlation across xy, yz, and xz planes discussed in SYM3D essentially corresponds to the inter-channel feature penetration problem we identify in our paper, though they observe it from a different angle. We detect this issue through visual inspection of feature maps (as shown in Figure 1(a,b)), while SYM3D quantifies it using correlation metrics.
>
> Inter-channel feature penetration causes similar values at the same UV positions across different feature maps, resulting in visually similar patterns, exactly what can be seen in our Figure 1(a,b). This phenomenon is also visible in the top-right panel of Figure 7 in SYM3D, where GET3D’s feature maps exhibit repetitive vertical lines at the same spatial locations when zoomed in. Numerically, this leads to high correlations between different feature maps.
>
> The difference lies in terminology: SYM3D does not explicitly refer to this as inter-channel feature penetration, but rather describes it as correlation.
>
> SYM3D attributes this issue to the use of single-view images during training, as opposed to multi-view data. While this is certainly true—more views provide richer geometric cues and reduce ambiguity—we also discuss this point in our paper (lines 65–71 and 224–227). From a more fundamental perspective, however, the issue arises due to the structural nature of convolutional networks, where different channels are prone to interfere with each other, especially in the absence of direct supervision. Our solution addresses this root cause directly by modifying the architecture via the unify-split strategy, which effectively eliminates inter-channel feature penetration at its source.
>
> In contrast, SYM3D uses view-wise spatial attention to alleviate the issue. As shown in the bottom-right panel of Figure 7 in SYM3D, this approach does reduce correlation to some extent. However, as illustrated in the middle-bottom panel, strong correlations still exist between certain planes, particularly Pyz and Pxz. We have also computed and visualized similarity matrices similar to those in SYM3D's Figure 7, and our results show significantly lower feature correlations across different planes. Please refer to our response to W4 & Q1 below for specific quantitative results.
>
> Moreover, in terms of implementation complexity, our unify-split strategy is simpler and more intuitive compared to SYM3D’s view-wise attention mechanism, while achieving a more complete resolution of the problem.
>
> We will include this detailed comparison in the final version of the manuscript to further clarify how our method improves upon previous approaches.
>
>
> **W3 & Q1: View-Specific Disentanglement Visualization**
>
> Thank you for pointing out the need for clearer visualization of view-specific disentanglement. Due to the NeurIPS’25 rebuttal policy, we are unable to include additional figures or external links in this response. However, we would like to provide a qualitative explanation based on Figure 4 in the paper.
>
> As designed in our hy-plane representation, the planar planes learn highly symmetric features, such as ears and shoulders, which are captured by the Pyz plane. On the other hand, the spherical plane captures asymmetric information such as the front face and back hair. This clear separation between symmetric and asymmetric features is a key aspect of the disentanglement achieved by our method.
>
> We will add more detailed visualizations and explanations of view-specific feature learning in the final version of the manuscript to further clarify this point.
>
>
> **W4 & Q1: Missing Quantitative Evaluation for Feature Penetration**
>
> Thank you for this valuable suggestion. We agree that a quantitative evaluation is essential to validate the effectiveness of our method in reducing feature penetration across channels.
>
> We recognize that the similarity matrix is a suitable metric for measuring inter-plane correlation and thus serves as an effective indicator of feature penetration. However, due to the NeurIPS’25 rebuttal policy, we are unable to include visualizations or figures in this response.
>
> Instead, we show the similarity matrix below and we will add it in the final manuscript. The results demonstrate that hy-plane(3+1) with the unify-split strategy achieves significantly lower correlations between different planes compared to hy-plane(3+1) without the unify-split strategy, indicating that the unify-split strategy effectively reduces inter-channel feature penetration.
>
> ```
> Similarity Matrix of hy-plane(3+1) with unify-split strategy:
> [[ 1.          0.13848868  0.19367712 -0.24672825]
>  [ 0.13848868  1.         -0.11731562  0.31794619]
>  [ 0.19367712 -0.11731562  1.         -0.47516028]
>  [-0.24672825  0.31794619 -0.47516028  1.        ]]
>
> Similarity Matrix of hy-plane(3+1) without unify-split strategy:
> [[1.         0.59808226 0.63007857 0.4739821 ]
>  [0.59808226 1.         0.6070948  0.71127384]
>  [0.63007857 0.6070948  1.         0.48041942]
>  [0.4739821  0.71127384 0.48041942 1.        ]]
> ```
>
> **W5 & Q3: Comparison in Single-view Setting**
>
> Thank you for raising this important point.
>
> We have conducted single-view reconstruction experiments using the same input image via PTI inversion. However, due to the NeurIPS’25 rebuttal policy, we are unable to include figures or external links in this response. Instead, we provide a qualitative discussion based on our results.
>
> The results show that all 3D-aware GAN-based methods (EG3D, PanoHead, SphereHead, and our HyPlaneHead) are capable of reconstructing highly detailed front-view images that closely match the input, as PTI finetunes the model parameters to fit the target image. However, significant differences emerge in the quality of back-view reconstructions.
>
> Specifically, EG3D and PanoHead often suffer from mirroring-face artifacts or lack sufficient expressiveness, leading to noticeable distortions in the back-of-the-head region. While SphereHead alleviates these issues, it tends to produce blurred and messy hair textures in the back view. In contrast, our method not only avoids such artifacts but also reconstructs clearer and more realistic details in the rear area.
>
> We will include the full set of single-view GAN inversion results and their corresponding analysis in the final version of the manuscript.

---

> ### Comment · Reviewer_abHE · 2025-08-07
>
> Dear Authors,
>
> Apologies that I could not respond earlier and thanks for the rebuttal. Yes, this rebuttal adressed most of my concerns and I am happy to keep my score.
>
> Please include the additional results in the rebuttal, especially W4, and relation to the previous work I mentioned (W1-2) in the revised paper.
>
> Best.

---

### Official Review · Reviewer_gW4q · 2025-07-02

**Clarity:** 3
**Significance:** 2
**Originality:** 2
**Rating:** 5
**Confidence:** 4

**Summary:**

The paper proposes HyplaneHead, a new re-parameterization of feature planes predicted by 3D GANs that operate with head portraits. The paper identifies two issues with commonly used output representations of 3D GANs which are tri-planes and spherical tri-planes, and proposes solutions to address these. First they identify the extensive interference between the planar output features in a tri-plane representation. This interference leads to the mixing of facial features across the different planes and often leads to strong artifacts when synthesizing novel views of the full head. For example, this interference can lead to frontal face features being visible when generating views of the back of the head. The authors argue that spherical tri-plane mitigate this issue to some level but unfortunately suffer from other drawbacks such as the poor spatial utilization of the square feature plane, leading to poor visual quality. To address these shortcomings, the paper proposes a Hyplane: a hybrid represent that has overcomes both drawbacks while leveraging the strengths of both tri-planes and spherical tri-planes. HyPlane is made up of two main contributions: i) The first is a lambertian azimuthal equal area (LAEA) projection scheme that maximizes the area occupied by spherical projection onto an rectangular plane. ii) The second contribution of the paper is a unify-split strategy where output planes in standard tri-plane representation are stitched onto a single plane to minimize cross-channel interference. Both improvements are analyzed through quantitative studies and lead to meaningful improvement in FID over the state-of-the-art.

**Questions:**

- Stitching the different planes into a single mosaic through the unify-split strategy implies that each plane now has a smaller effective resolution on the feature plane assuming the spatial resolution of the generator is kept the same as the baseline. The authors discuss a couple of different splitting strategies (even vs. biased). Does the smaller effective resolution for a dominating plane have any adverse effects on the quality of the results?

- The paper discusses HyPlane (2+2) as a generalized solution that could extend to general scenes. Did the authors try training a 3D GAN on general scenes as well?

**Ethical Concerns:**

["NO or VERY MINOR ethics concerns only"]

**Final Justification:**

My concerns were addressed by the authors in the rebuttal. As long as the promised results are included in the final version of the paper, I am happy to recommend acceptance.

**Limitations:**

The paper suffers from the standard limitations of 3D GANs that are trained with 2D image datasets including the flickering artifacts when generating novel views and its inability to generalize to diverse, out-of-distribution hairstyles, identities and so on.

The paper discusses both limitations.

**Paper Formatting Concerns:**

N/A.

**Quality:**

3

**Strengths And Weaknesses:**

*Strengths*
- The paper addresses a practical problem plaguing 3D GANs and proposes two working yet simple solutions to address them. I like the focus of the paper.
- Both contributions of the paper (the unify-split strategy and the LAEA projection) are shown to offer meaningful improvements in FID.

*Weakness*
- The primary criticism I have is around the presentation of results. I would have liked to see comparisons of the extracted surface for different methods. Specifically it would have been interesting to see what effect the two proposed improvements have on the underlying 3D surface.
- The qualitative results are presented in a manner that makes it hard to compare the different methods. It would help a lot if the paper had text labels next to the different results (Chan et al., Li et al., Ours etc.). Currently one is forced to parse a large amount of text to understand which result is what. The same is true for most figures in the supplemental as well. The figure caption simply states “comparison” and one is left with no visual guidance to parse the figures.
- While the paper addresses practical problems with 3D GANs used for generating 360 degree head portraits, I wonder whether its contributions are of sufficient novelty for NeurIPS.

---

> ### Author Rebuttal · Authors · 2025-07-28
>
> Thanks for your insightful review! Here are our responses to your concerns.
>
> **W1: Surface Comparisons**
>
> We apologize that, due to the NeurIPS’25 rebuttal policy, we are unable to include additional figures or external links in this response. However, we would like to provide a qualitative discussion based on the geometry visualizations presented in the paper and some results not included in the main submission.
>
> *All surface/shape/geometry comparisons mentioned below were extracted from tri-plane-like representations using the Marching Cubes algorithm and saved as .mrc files for visualization with ChimeraX software. This is consistent with the extraction methods used in EG3D, PanoHead, and SphereHead.*
>
> For the tri-plane representation, originally proposed in EG3D (as shown in Figure 5(a)), one can observe a face-like artifact appearing on the back of the head when viewing at 90°, 135°, and 180° angles. In the corresponding 3D surface, this manifests as an unnatural facial-shaped protrusion. This issue is commonly referred to as the "mirroring-face artifact," as discussed in the SphereHead paper.
>
> For the tri-grid representation, introduced in PanoHead, the addition of parallel planes enhances representational capacity, which alleviates the mirroring problem to some extent. However, a similar face-like artifact still appears on the back of the head. From the extracted surface, this artifact is less pronounced and more flattened compared to the tri-plane case. Nevertheless, the issue is not fully resolved. Moreover, both tri-grid and tri-plane always suffer from high-frequency grid-like noise patterns, which are visible as horizontal and vertical stripes, in certain regions of the extracted surface, especially around the back of the head. This is caused by the use of Cartesian coordinate projection for feature querying.
>
> For the single spherical tri-plane representation, as proposed in SphereHead (Figure 2(f)), the extracted surface appears smoother but lacks fine details. This is primarily due to the uneven expressiveness across different spatial locations and the underutilization of the square feature map of the spherical plane. Additionally, polar artifacts appear on left and right sides and seam artifacts are visible on the back of the head.
>
> For the dual spherical tri-plane representation, also from SphereHead (Figure 5(e)), while polar artifacts and seam artifacts are disappeared, the extracted surface looks even smoother. This is largely because each spherical plane uses its weakly expressive equatorial region to cover the other’s strongly expressive poles, leading to suboptimal overall detail and quality.
>
> In contrast, our hy-plane representation (Figure 1(d) and Figure 5(j)) achieves a surface that retains rich geometric detail while eliminating the grid-like noise patterns seen in tri-grid/tri-plane methods. It also completely resolves the mirroring-face artifact, just as the spherical tri-plane does. We further observe that the introduction of the unify-split strategy eliminates cross-channel feature penetration artifacts. For example, the abnormal horizontal streaks often seen in hair regions in tri-plane/tri-grid methods (caused by features from Pyz leaking into Pxy) no longer appear in our method.
>
> Additionally, the integration of the near-equal-area warping strategy (LAEA projection + elliptical grid mapping) significantly improves the clarity of fine details in the extracted surface. This is due to the more efficient utilization of the square feature map for the spherical plane.
>
> We will include detailed surface comparison figures in the final version of the manuscript.
>
> **W2: Lack of Clear Visual Labels in Qualitative Comparisons**
>
> We sincerely thank the reviewer for this valuable feedback. We agree that clearer visual labels would greatly improve the readability and comparison of our qualitative results. In the final version of the paper, we will revise all figures (including those in the supplementary material) to include explicit text labels next to each method’s output for better clarity and ease of comparison.
>
> **W3 & Q2: Novelty and Generalization to Beyond Full-Head Scenarios**
>
> We appreciate the reviewer’s thoughtful question regarding the novelty and general applicability of our method.
>
> While the current work focuses on full-head generation due to data availability and practical relevance, we have also conducted preliminary experiments on other objects to evaluate the generalization potential of hy-plane.
>
> Specifically, we collected multiple categories (e.g., cars, motorcycles, furniture, fruits, and houses) from public 3D model datasets (e.g. Objaverse, OmniObject3D and ShapeNet), and trained our model on each category. The preliminary results show that hy-plane achieves performance comparable to the standard tri-plane representation, particularly for symmetric objects such as cars and furniture. For asymmetric objects like fruits or houses, our method can produce more accurate asymmetric regions and generates fewer mirroring artifacts.
>
> For example, when generating a house, although some houses exhibit symmetry in shape and window placement, they typically do not have two symmetric doors, i.e. one on each side. However, using a standard tri-plane tends to generate either two unrealistic "doors" or door-like artifacts on the opposite side. In contrast, our hy-plane naturally captures the asymmetry and typically generates only one realistic door, without any mirror-like artifacts on the other side.
>
> Exploring the application of hy-plane to more general objects and scenes is an interesting and valuable direction. Due to time and data constraints during the rebuttal phase, we have not yet extended our experiments to more general scenarios such as object generation across arbitrary categories. We will include these additional experiments and discussions in the final version of the manuscript.
>
> **Q1: Resolution in Unify-Split Strategy**
>
> In our experiments, we used a larger unified feature map of size 512×512 so that after splitting, each individual feature map would have the same resolution (256×256) as those used in tri-plane, tri-grid, and spherical tri-plane methods. Although this increases the model’s parameter count to some extent, our experiments show that the performance improvement is not due to the increased number of parameters.
>
> *Due to space limitations, please refer to the Representation Parameters and Total Learnable Parameters columns in the table provided in our response to Reviewer Xp1z (Q1 & Q2) for detailed information.*
>
> As shown in Table 1 (rows 8, 9, and 10), even when the feature maps of tri-plane, tri-grid, and spherical tri-plane are also upscaled to 512×512, their performance does not improve compared to the 256×256 version. This indicates that the performance gain in our method is attributed to the novel representation design rather than an increase in model capacity.
> Besides, the same observation can be found in Table 2 of OrthoPlanes [1]. The first and second rows compare the same EG3D-based setup with only one difference: the former uses a 256×256 feature map, while the latter uses a 512×512 feature map. However, the FID score of the latter is actually higher than that of the former, demonstrating that simply increasing the feature map size does not necessarily lead to better performance.
>
> Furthermore, although we use a larger unified feature map (512×512), it has little impact on inference speed or GPU memory usage. Therefore, we believe the comparison between different methods remains fair and meaningful.
>
> [1] OrthoPlanes: A Novel Representation for Better 3D-Awareness of GANs

---

> > ### Comment · Reviewer_gW4q · 2025-08-05
> >
> > Thanks a lot for the detailed response. You've addressed most of my concerns with your rebuttal.
> >
> > Please make sure to include the additional results you've promised in the final version of the paper; especially the surface comparisons, better labelling of figures and some preliminary results of training on general scenes. These will significantly improve the interest and readability of your work :)

---

### Official Review · Reviewer_Xp1z · 2025-07-03

**Clarity:** 3
**Significance:** 4
**Originality:** 3
**Rating:** 5
**Confidence:** 3

**Summary:**

HyPlaneHead introduces a new hybrid-plane (hy-plane) representation for 3D-aware GANs, improving full-head image synthesis by reducing artifacts like feature entanglement and mirroring. The proposed method combines planar and spherical planes, using innovations like near-equal-area warping and a unify-split strategy, achieving state-of-the-art performance. In addition,  the single-channel unified feature map design mitigate the feature penetration problem, which is a significant contribution of this work.

**Questions:**

**Q1**: As mentioned in the weakness part, could the authors share the influence of each component on the training and inference speed?

**Q2**: The proposed "Unify-Split Strategy" did managed to mitigate the feature penetration problem, but might introduce additional GPU memory usage. Could this method be extended to higher resolution image generation without a super-resolution module? More training details will be highly appreciated.

**Q3**: Could the authors provide some single-view GAN Inversion results with PTI? The inversion results might be useful for some other real-world application and could better showcase the model capacity.

**Ethical Concerns:**

["NO or VERY MINOR ethics concerns only"]

**Final Justification:**

Issuses resolved: The author provided a detailed analysis regarding the computational cost and proposed a potential solution for extending the method to higher resolutions.

Issuses unresolved: Due to the rebuttal policy, the author was unable to provide experimental results on inversion. While this is unfortunate, it is also understandable.

I believe the issues unresolved should not detract from the fact that this is an innovative work, especially considering its significant contribution to alleviating mirroring-face artifacts.

**Limitations:**

Yes

**Quality:**

3

**Strengths And Weaknesses:**

Strength:

**Novelty and Innovation**: The hy-plane representation is a novel approach, addressing limitations in existing tri-plane methods. It leverages planar planes for symmetry (e.g., capturing left-right symmetry in heads) and spherical planes for anisotropy, potentially setting a new standard for 3D-aware GANs.

**Technical Advancements**: The near-equal-area warping and unify-split strategy are innovative, with potential applications beyond full-head synthesis. For instance, the unify-split strategy could reduce feature entanglement in other 3D object synthesis tasks, enhancing model expressiveness.

**Empirical Support**: The paper provides comprehensive experiments, including ablation studies and comparisons with state-of-the-art methods (e.g., Chan et al. 2022, An et al. 2023, Li et al. 2024). The superior FID scores and reduced artifacts in visualizations support the claimed improvements.


**Weakness:**

**Model Complexity**: My major concern lies in the model complexity. The proposed method introduce several novel component. Despite the impressive results, these components might introduce additional computational overhead, but the authors did not discuss the training and inference speed. It would be great if the authors could share the influence of each component on the speed.

---

> ### Author Rebuttal · Authors · 2025-07-25
>
> Thanks for your insightful review! Here are our responses to your concerns.
>
> **Q1 & Q2: Training/Inference Speed & VRAM Usage**
>
> To provide a comprehensive comparison of training and inference speed, as well as VRAM usage across different methods and HyPlaneHead configurations, we conducted measurements for each experiment listed in Table 1 of the paper. The results are as follows.
>
> | No. | Representation              | Unify-Split     | Wrapping | Representation Parameters | Total Learnable Parameters | Training Speed (sec/kimg) | Inference Speed (ms/image) | Training VRAM Usage (MiB) | Inference VRAM Usage (MiB) |
> |-----|-----------------------------|-----------------|----------|---------------------------|----------------------------|-------------------------|---------------------------|---------------------------|----------------------------|
> | 1   | Tri-plane                   | -               | -        | 3x256x256                 | 53,174,956                 | 180.29                  | 42.06                     | 6436                   | 1103                    |
> | 2   | Tri-plane                   | evenly split    | -        | 1x512x512                 | 53,230,222                 | 197.89                  | 44.17                     | 7670                   | 1149                    |
> | 3   | Spherical Tri-plane         | -               | -        | 3x256x256                 | 54,713,868                 | 222.84                  | 54.99                     | 8048                   | 1481                    |
> | 4   | Spherical Tri-plane         | evenly split    | -        | 1x512x512                 | 53,234,479                 | 198.57                  | 48.19                     | 7692                   | 1097                    |
> | 5   | Dual Spherical Tri-plane    | -               | -        | 6x256x256                 | 54,713,868                 | 245.79                  | 54.87                     | 9296                   | 1481                    |
> | 6   | Dual Spherical Tri-plane \* | -               | -        | 6x256x256                 | 53,462,509                 | 196.88                  | 50.42                     | 6810                   | 1235                    |
> | 7   | Tri-grid                    | -               | -        | 9x256x256                 | 53,741,548                 | 198.59                  | 45.19                     | 6836                   | 1245                    |
> | 8   | Tri-plane 512^2             | -               | -        | 3x512x512                 | 53,423,246                 | 181.00                  | 47.77                     | 6450                   | 1333                    |
> | 9   | Spherical Tri-plane 512^2   | -               | -        | 6x512x512                 | 54,962,158                 | 182.28                  | 59.91                     | 6744                   | 1103                    |
> | 10  | Tri-grid 512^2              | -               | -        | 9x512x512                 | 54,002,318                 | 266.13                  | 58.79                     | 8642                   | 2463                    |
> | 11  | Hy-plane (3+1)              | -               | -        | 4x256x256                 | 53,269,388                 | 190.40                  | 43.64                     | 6966                   | 1095                    |
> | 12  | Hy-plane (3+1)              | evenly split    | -        | 1x512x512                 | 53,230,222                 | 206.54                  | 46.27                     | 7818                   | 1277                    |
> | 13  | Hy-plane (3+1)              | evenly split    | yes      | 1x512x512                 | 53,230,222                 | 207.21                  | 47.89                     | 7432                   | 1205                   |
> | 14  | Hy-plane (3+1)              | area-bias split | yes      | 1x512x512                 | 53,230,222                 | 226.31                  | 49.61                     | 7558                   | 1321                    |
> | 15  | Hy-plane (2+2)              | evenly split    | yes      | 1x512x512                 | 53,230,222                 | 212.27                  | 49.81                     | 7780                   | 1215                    |
> | 16  | Hy-plane (2+2)              | area-bias split | yes      | 1x512x512                 | 53,230,222                 | 219.73                  | 51.74                     | 8012                   | 1255                    |
>
> To provide a comprehensive and fair comparison, we clarify the definitions of the metrics used in our evaluation. Representation Parameters refer to the number of parameters in the feature maps of different tri-plane-like representations (e.g., tri-plane, spherical plane, hy-plane, etc.). Total Learnable Parameters denote the total number of trainable parameters in the entire model architecture, such as EG3D, PanoHead, SphereHead, and HyPlaneHead. Training Speed and Training VRAM Usage are measured on a single V100 GPU with a batch size of 2, representing the average time and memory consumption required to train 1,000 images. Similarly, Inference Speed and Inference VRAM Usage are evaluated under the same hardware setup but with a batch size of 1, by generating 100 images and computing the average time and memory cost per image.
>
> From the statistics in the table, we can observe that different representations vary significantly in terms of feature plane parameter count. The tri-plane uses the fewest parameters (3 × 256 × 256 floating-point values), while the tri-grid with 512×512 resolution uses the most (9 × 512 × 512). Our proposed hy-plane uses 1 × 512 × 512 floating-point values, which is only 1.33 times the number used by the tri-plane. However, the overall difference in total learnable parameters across models is relatively small. The minor variations are mainly due to differences in the final convolutional layer configuration of StyleGAN2, which depends on how each representation is generated.
>
> Notably, the total learnable parameters for entries No. 3 and No. 5 are identical. This is because in experiment No. 3, we did not modify the model code at all, which means only the rendering pipeline was adjusted to use one spherical tri-plane. We reported the actual parameter count to remain consistent with our experimental setup.
>
> In terms of training and inference speed, as well as VRAM usage, there are some differences among the experiments. Overall, however, the additional computational overhead introduced by our innovations is relatively small.
>
> Replacing the tri-plane with hy-plane introduces:
> * +5.5% training time,
> * +3.8% inference time,
> * +8.8% training VRAM,
> * −0.8% inference VRAM.
>
> Adding the evenly split strategy further increases the cost by:
> * +8.4% training time,
> * +6.9% inference time,
> * +12.2% training VRAM,
> * +16.6% inference VRAM.
>
> When combining with the near-equal-area projection, the overhead becomes:
> * +0.5% training time,
> * +3.5% inference time,
> * −5% training VRAM,
> * −5.7% inference VRAM.
>
> Please note that in some cases, VRAM usage may actually decrease, likely due to memory fragmentation causing inaccuracies in the nvidia-smi measurement.
>
> In summary, while our method does introduce some computational and memory overhead, the increase is relatively modest and justifiable given the significant improvements in disentanglement and reconstruction quality.
>
> In addition, regarding Q2, which asks "Could this method be extended to higher-resolution image generation without a super-resolution module?" This is indeed a very insightful question. Current 3D-aware GANs typically rely on super-resolution modules to enhance the rendering resolution. However, it is possible that alternative strategies could achieve similar results. For example, Tri2-Plane [1] proposes a method of incorporating a feature pyramid into the tri-plane structure to improve fine-grained detail generation. Whether such an approach can fully replace the widely used super-resolution modules remains an open and interesting question. We consider this a promising direction for future work and will explore it in more depth in subsequent research.
>
> [1] Tri2-Plane: Thinking Head Avatar via Feature Pyramid, 2024
>
>
> **Q3: Single-View GAN Inversion Results:**
>
> We have conducted single-view reconstruction experiments using the same input image via PTI inversion. However, due to the NeurIPS’25 rebuttal policy, we are unable to include figures or external links in this response. Instead, we provide a qualitative discussion based on our results.
>
> The results show that all 3D-aware GAN-based methods (EG3D, PanoHead, SphereHead, and our HyPlaneHead) are capable of reconstructing highly detailed front-view images that closely match the input, as PTI finetunes the model parameters to fit the target image. However, significant differences emerge in the quality of back-view reconstructions.
>
> Specifically, EG3D and PanoHead often suffer from mirroring-face artifacts or lack sufficient expressiveness, leading to noticeable distortions in the back-of-the-head region. While SphereHead alleviates these issues, it tends to produce blurred and messy hair textures in the back view. In contrast, our method not only avoids such artifacts but also reconstructs clearer and more realistic details in the rear area.
>
> We will include the full set of single-view GAN inversion results and their corresponding analysis in the final version of the manuscript.

---

### Note · Authors · 2025-08-11

Thank you to all the reviewers and the Area Chair for your valuable feedback, suggestions, and insightful discussions. We will incorporate these comments and related results into the final version of the paper and supplementary material accordingly.

---

### Decision · Program_Chairs · 2025-09-17

**Decision:**

Accept (poster)

**Comment:**

All four reviewers rated the paper positively, with concerns addressed convincingly in the rebuttal. HyPlaneHead’s ​hybrid-plane representation​ effectively eliminates mirroring artifacts in 3D head synthesis by combining planar/spherical planes with innovations like ​near-equal-area warping​ and a ​unify-split strategy. Quantitative results show significant FID improvements over SOTA methods while rebuttal responses thoroughly addressed concerns in computational over head, generalizability, ​inversion quality​ etc. The rebuttal is successful and reviewers' concerns are mostly resolved.